EMBO
Molecular Medicine

# Counteracting the effects of TNF receptor-1 has therapeutic potential in Alzheimer's disease

Sophie Steeland[1,2,†], Nina Gorlé[1,2,†], Charysse Vandendriessche[1,2,†], Sriram Balusu[1,2], Marjana Brkic[1,2,3], Caroline Van Cauwenberghe[1,2], Griet Van Imschoot[1,2], Elien Van Wonterghem[1,2], Riet De Rycke[1,2], Anneke Kremer[1,2,4], Saskia Lippens[1,2,4], Edward Stopa[5,6], Conrad E Johanson[6], Claude Libert[1,2] & Roosmarijn E Vandenbroucke[1,2,*] 

## Abstract

Alzheimer's disease (AD) is the most common form of dementia, and neuroinflammation is an important hallmark of the pathogenesis. Tumor necrosis factor (TNF) might be detrimental in AD, though the results coming from clinical trials on anti-TNF inhibitors are inconclusive. TNFR1, one of the TNF signaling receptors, contributes to the pathogenesis of AD by mediating neuronal cell death. The blood–cerebrospinal fluid (CSF) barrier consists of a monolayer of choroid plexus epithelial (CPE) cells, and AD is associated with changes in CPE cell morphology. Here, we report that TNF is the main inflammatory upstream mediator in choroid plexus tissue in AD patients. This was confirmed in two murine AD models: transgenic APP/PS1 mice and intracerebroventricular (icv) AβO injection. TNFR1 contributes to the morphological damage of CPE cells in AD, and TNFR1 abrogation reduces brain inflammation and prevents blood–CSF barrier impairment. In APP/PS1 transgenic mice, TNFR1 deficiency ameliorated amyloidosis. Ultimately, genetic and pharmacological blockage of TNFR1 rescued from the induced cognitive impairments. Our data indicate that TNFR1 is a promising therapeutic target for AD treatment.

**Keywords** Alzheimer's disease; blood-CSF barrier; choroid plexus; therapy; TNF receptor 1

**Subject Categories** Immunology; Neuroscience; Pharmacology & Drug Discovery

## Introduction

Alzheimer's disease (AD) is the leading cause of dementia, with a prevalence of about 10% after the age of 65 (Hebert *et al*, 2013; Prince *et al*, 2015). AD is a chronic and progressive neurodegenerative disease characterized by brain atrophy and accumulation of amyloid β (Aβ) plaques in the brain parenchyma and neurofibrillary tangles in the neurons (Querfurth & Laferla, 2010). The importance of soluble Aβ oligomers (AβO)—the intermediate form between monomeric Aβ and Aβ fibrils—has gained increasing attention as the major toxic species driving the neuroinflammation associated with Alzheimer's disease rather than the Aβ plaques themselves (Nitta *et al*, 1997; Brkic *et al*, 2015b; Ferreira *et al*, 2015; Viola & Klein, 2015). Neuroinflammation is an important hallmark of brain pathology (Akiyama *et al*, 2000), and both acute and chronic inflammation are associated with cognitive decline (Holmes *et al*, 2009).

Tumor necrosis factor (TNF) is a pleiotropic pro-inflammatory cytokine and a key mediator in many inflammatory disorders, such as inflammatory bowel disease (IBD) and rheumatoid arthritis (RA), as well as neurodegenerative diseases such as Parkinson's disease and AD (Probert, 2015). Besides its inflammatory role, TNF functions in the brain as a gliotransmitter secreted by neurons and glial cells. It regulates synaptic communication between neurons and thereby regulates brain function in health and in disease (Stellwagen & Malenka, 2006; McCoy & Tansey, 2008). It has been proposed that the effects of TNF on synapses are associated with the synaptic dysfunction that plays a central role in AD, and particularly in cognitive decline (Tobinick, 2009). Clinical evidence for the involvement of TNF in AD comes from the co-localization of TNF with Aβ plaques and the elevated TNF levels in the plasma and cerebrospinal fluid (CSF) of AD patients, both of which have been associated with clinical deterioration (Fillit *et al*, 1991; Dickson, 1997; Paganelli *et al*, 2002; Tarkowski *et al*, 2003a,b; Yasutake *et al*,

1 VIB Center for Inflammation Research, Ghent, Belgium
2 Department of Biomedical Molecular Biology, Ghent University, Ghent, Belgium
3 Department of Neurobiology, Institute for Biological Research, University of Belgrade, Belgrade, Republic of Serbia
4 VIB BioImaging Core, Ghent, Belgium
5 Department of Pathology, Rhode Island Hospital, Providence, Rhode Island, USA
6 Department of Neurosurgery, Warren Alpert Medical School of Brown University, Providence, Rhode Island, USA
*Corresponding author. Tel: +32 (0)9 331 35 87; Fax: +32 (0)9 221 76 73; E-mail: roosmarijn.vandenbroucke@irc.vib-ugent.be
†These authors contributed equally to this work

2006; Swardfager *et al*, 2010). In addition, it has been speculated that TNF contributes to amyloidogenesis in AD (Blasko *et al*, 1999) and that chronic neuronal TNF overexpression promotes brain inflammation, which is detrimental to neuronal viability (Janelsins *et al*, 2008). Epidemiological studies indicate that AD is less prevalent in patients on anti-TNF blockers for rheumatoid arthritis (Chou *et al*, 2016), but clinical trials using anti-TNF blockers in AD patients have been inconclusive (Tobinick *et al*, 2006; Tobinick & Gross, 2008). Furthermore, studies in mice raise questions about the use of pan-TNF inhibitors in AD (Montgomery *et al*, 2011). TNF can signal through two distinct receptors: TNF receptor 1 (TNFR1) and TNFR2. Activation of TNFR1 leads to pro-inflammatory and neurotoxic activities, whereas triggering TNFR2 leads to immunomodulation and has neuroprotective roles. It has been shown that TNFR1 protein levels and binding affinity in AD brains are increased, while TNFR2 levels and binding affinity are decreased compared to non-demented patients (Zhao *et al*, 2003; Cheng *et al*, 2010). Additionally, Li *et al* (2004) reported that in the AD brain, TNFR1 mediates Aβ-induced neuronal cell death. Moreover, it is believed that TNFR1 plays a role in amyloidogenesis by regulating β-secretase (BACE1), one of the enzymes important for processing amyloid precursor protein (APP). In contrast, inhibition of TNFR2 increased Aβ toxicity *in vitro* (Shen *et al*, 1997), showing the importance of preserving TNFR2 so that it can perform its functions in the neuroprotective pathways. Moreover, when transgenic APP23 mice (a model of AD) were backcrossed in a TNFR2-deficient background, they displayed exacerbated AD pathology compared to APP23 mice with a functional TNFR2 gene (Jiang *et al*, 2014). Also, selective inhibition of neuronal TNFR2 enhanced the Aβ- and Tau-related pathologic features of AD and diminished the activation of microglia, which is needed for clearance of Aβ (Montgomery *et al*, 2013). These observations indicate that while TNFR1 plays detrimental roles in AD, TNFR2 needs to be preserved in order to counteract the Aβ-mediated pathology. This requires selective targeting of the TNF pathway.

Numerous studies indicate that neurovascular dysfunction and perturbation of the central nervous system (CNS) barriers contribute to the onset and progression of AD (Zenaro *et al*, 2017). The CNS is strictly protected against external toxic insults by two important barriers: the blood–brain barrier (BBB) and the blood–CSF barrier. The choroid plexus is a highly vascularized specialized structure in the ventricles of the brain consisting of a monolayer of cuboidal choroid plexus epithelial (CPE) cells. These cells produce CSF and form the blood–CSF barrier. The apical side of CPE cells faces the CSF and contains numerous microvilli. In contrast to the BBB, which is formed by tightly connected endothelial cells, the capillaries of the choroid plexus are highly fenestrated, whereas the CPE cells are firmly interconnected, separating blood from CSF (Balusu *et al*, 2016a; Gorle *et al*, 2016; Vandenbroucke, 2016). Consequently, the basal side of the choroid plexus is in contact with molecules in the blood, but the blood–CSF barrier prevents their passage to the brain parenchyma. Only active transport of specific molecules across the barrier is allowed. Accordingly, the choroid plexus system is actively involved in physiological processes and is essential for maintaining brain homeostasis (Vandenbroucke, 2016). The choroid plexus is also an important sensor of inflammatory stimuli in the blood (Vandenbroucke *et al*, 2012; Balusu *et al*, 2016a,b), and it may act as a selective gateway to the brain for the trafficking

of immune cells (Demeestere *et al*, 2015). However, in AD this gateway system is altered, resulting in suboptimal recruitment of inflammation-resolving immune cells to the brain (Baruch *et al*, 2015). In addition, numerous changes in choroid plexus morphology and function in AD have been described (Serot *et al*, 2000, 2003, 2012; Brkic *et al*, 2015b). Additionally, there is a clear inflammatory signature in the choroid plexus of transgenic AD mice but not in age-matched WT littermates (Mesquita *et al*, 2015).

We recently found that intracerebroventricular (icv) injection of Aβ$_{1–42}$ oligomers (AβO) induces choroid plexus inflammation and loss of the blood–CSF barrier integrity, which was linked to increased activity of matrix metalloproteinases (MMPs) (Brkic *et al*, 2015b). In our present study, we show that TNF/TNFR1 signaling is the main axis contributing to CNS neuroinflammation in transgenic AD mice and upon AβO injection in WT mice. We demonstrate that TNFR1 deficiency protects against structural and functional impairment of the choroid plexus and eventually rescues cognitive decline. Furthermore, we provide proof of concept that pharmacological inhibition of TNFR1 can prevent the AβO-associated cognitive decline.

# Results

## TNF is the top upstream regulatory cytokine in the choroid plexus of late-stage AD patients

Numerous papers have highlighted the importance of the choroid plexus in AD (Serot *et al*, 2000, 2003, 2012; Brkic *et al*, 2015b). In the current study, we performed a microarray analysis to identify genes that are differentially expressed in human choroid plexus tissue obtained post-mortem from late-stage AD patients and from healthy controls. The ages of these subjects were 79 ± 2.9 and 58 ± 5.6 years, respectively, and thus were slightly different. Before analysis, the quality of the microarray data was stringently checked. Samples that did not pass hybrid quality control were excluded from analysis.

We analyzed differentially expressed genes in the choroid plexus of AD patients by using ingenuity pathway analysis (IPA; Fig 1). IPA recognizes the cascade of upstream transcriptional regulators that explains the differential expression and thereby identifies changes in biological activities. The *z*-score was determined for several upstream transcriptional regulators, with focus on upstream cytokines. A *z*-score > 0 indicates that the upstream transcriptional regulator has significantly more "activated" than "inhibited" predictions, and *vice versa* for a *z*-score < 0. As shown in Fig 1A, "upstream regulator analysis" in IPA revealed that *TNF* and *IL1β* were the most prevalent activated upstream transcriptional regulators of cytokines (*z*-score of 6.98 and 6.53, respectively). Other pro-inflammatory genes (such as *IL6* and *IFNG*) were also found upstream of the differentially expressed genes (*z*-score 5.38 and 4.92, respectively). Strikingly, *TNF* itself was not differentially expressed in the choroid plexus of late-stage AD patients (data not shown). Therefore, we looked for genes downstream of *TNF* that are differentially expressed in the choroid plexus. Figure 1B shows that the expression of 157 TNF-dependent genes was altered in the choroid plexus of late-stage AD patients, and most of them were upregulated (indicated in red). Therefore, TNF also directly induces

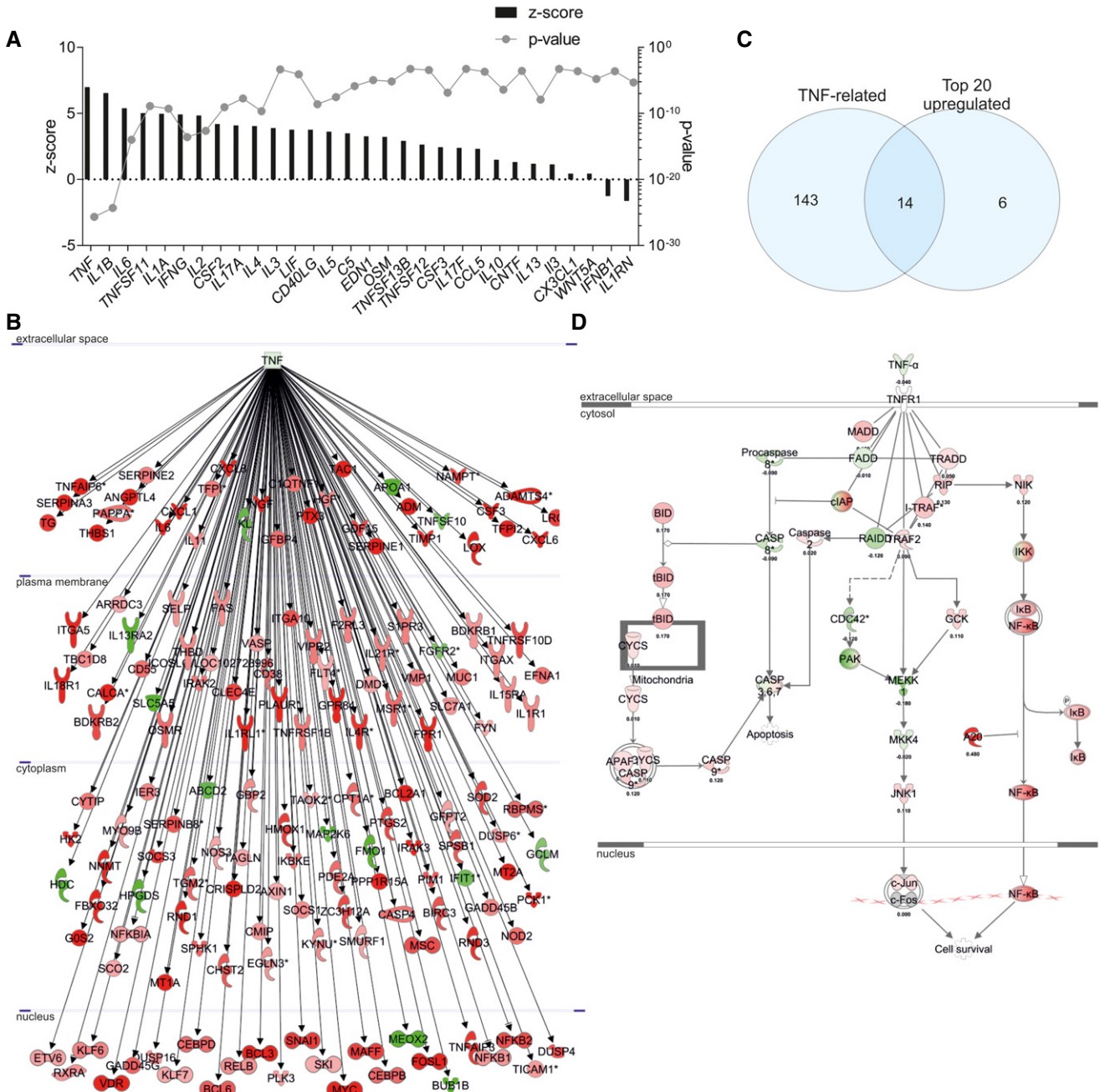

**Figure 1.   Ingenuity pathway analysis analysis of microarray results of human choroid plexus of late-stage AD patients.**

Ingenuity pathway analysis (IPA) was used to identify the pathways of differentially expressed genes in the choroid plexus of patients with late-stage Alzheimer's disease (AD) compared to age-matched healthy controls.

A   Identification of upstream cytokine mediators in the choroid plexus of AD patients. The upstream mediators are ranked according to their *z*-score (left axis) and *P*-value (right axis).
B   Network of differentially expressed genes downstream of *TNF* in the choroid plexus of late-stage AD patients compared to age-matched healthy controls (*red = upregulated, green = downregulated*).
C   Venn diagram of the top 20 differentially expressed genes and the differentially expressed genes downstream TNF.
D   Overlay of the dataset on the canonical pathway of the TNF/TNFR1 [*red = upregulated* (with log ratio), *green = downregulated* (with log ratio)].

pro-inflammatory cytokines such as *IL6* and members of the IL1-signaling pathway such as IL1 receptor 1 (*IL1R1*). Further, analysis of the overlap between the top 20 upregulated genes in the dataset and the differentially expressed genes downstream of *TNF* (Fig 1C) revealed that the TNF-dependent changes in gene expression were very pronounced.

The pro-inflammatory signaling pathway of TNF is mediated by signaling through TNFR1 and generally leads to the induction of globally activated transcription factors such as NF-κB (Wajant & Scheurich, 2011; Sedger & McDermott, 2014). This is also clear in our dataset (red in Fig 1D). Though *TNFRSF1a* was not upregulated in the choroid plexus of late-stage AD patients, NF-κB activity was clearly upregulated.

### NF-κB-dependent genes are upregulated in transgenic AD mice and upon AβO injection in wild-type mice

Next, we studied the expression of several NF-κB-induced genes in a transgenic mouse model for amyloidogenesis (APP/PS1 mice) (Radde *et al*, 2006). We first determined the induction of several NF-κB-dependent genes in the choroid plexus and hippocampus,

which is the key brain region involved in the formation of new memory and the first region that exhibits neurodegeneration in AD (Hollands *et al*, 2016). Figure 2A and B compares APP/PS1$^{tg/wt}$ mice with age-matched non-transgenic littermates (APP/PS1$^{wt/wt}$).

Analysis of the choroid plexus (Fig 2A) and hippocampus (Fig 2B) of late-stage APP/PS1$^{tg/wt}$ mice showed that several pro-inflammatory genes were significantly induced. The most pronounced upregulated genes in the choroid plexus of transgenic mice were of *Il6, Nos2, Lcn2, Mmp8,* and *Mmp13*. Interestingly, in agreement with the data obtained from human AD patients, *Tnf* and *Tnfrsf1a* were not differentially expressed in the choroid plexus of APP/PS1$^{tg/wt}$ mice. However, in the hippocampus, *Tnf* and *Tnfrsf1a*, as well as *Cxcl9* and *Lcn2*, were significantly induced.

The APP/PS1 transgenic mouse model mimics important hallmarks of human AD pathology. To study the direct effects of Aβ, we

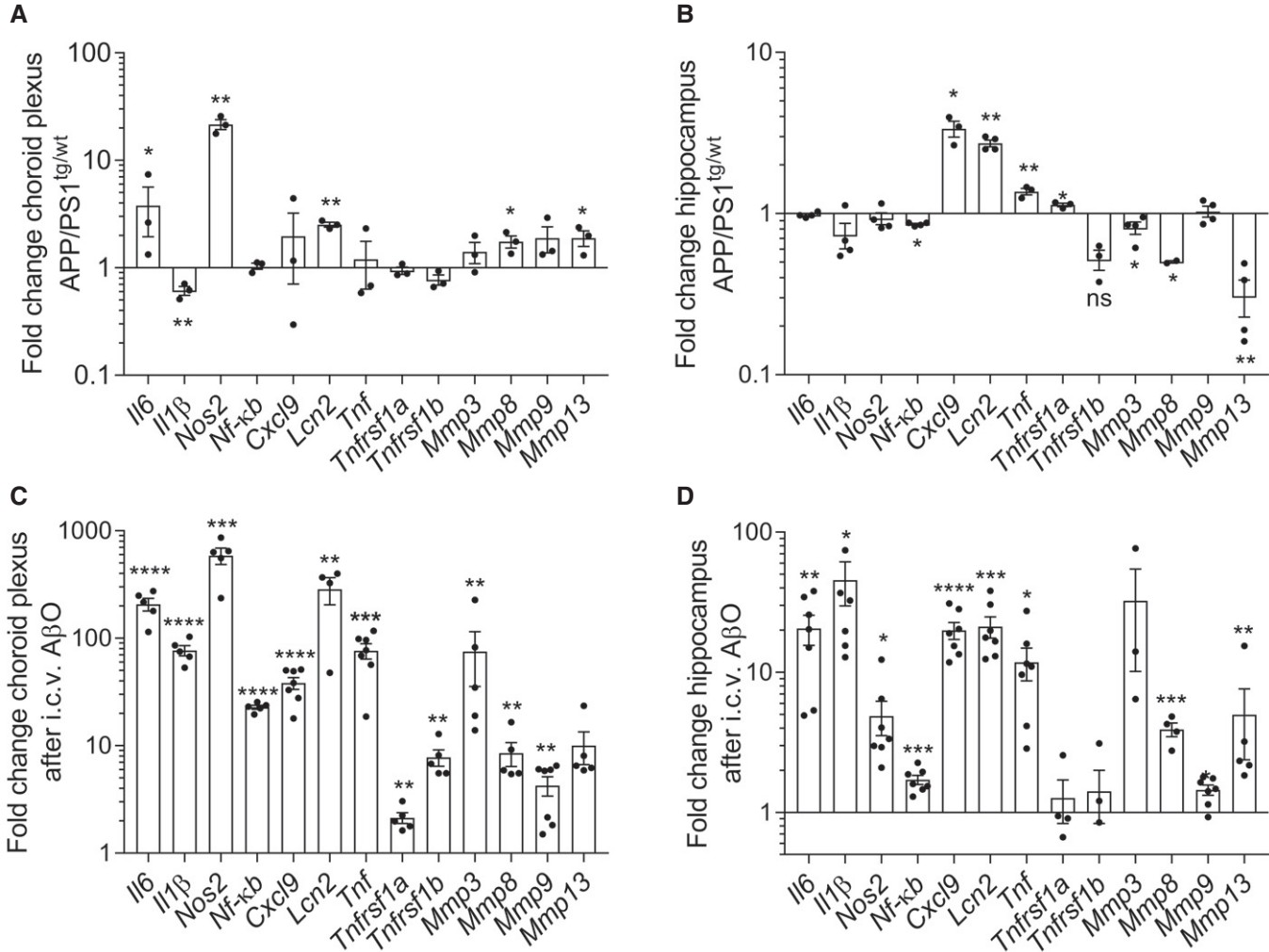

**Figure 2. Analysis of NF-κB-dependent genes in APP/PS1 mice and in Aβ$_{1–42}$ oligomer (AβO)-injected wild-type (WT) mice.**

A–D  Fold change in mRNA gene expression in the choroid plexus and hippocampus determined by qPCR of late-stage APP/PS1$^{tg/wt}$ mice compared with age-matched control mice (A, B) (*n* = 3–4/group) and of C57BL6/J WT mice 6 h after intracerebroventricular (icv) injection with AβO (1 μg/ml) compared to icv injection with scrambled peptide (C, D) (*n* = 5–7/group).

Data information: Bars represent mean ± SEM. qPCR was normalized to stable housekeeping genes, determined by GeNorm. Statistics between control mice and AD mice were done with an unpaired *t*-test, *0.01 ≤ *P* < 0.05; **0.001 ≤ *P* < 0.01; ***0.001 ≤ *P* < 0.0001, ****P < 0.0001.

injected 5 μl of 1 μg/ml oligomerized Aβ$_{1-42}$ (AβO) in the cerebral ventricles (icv) of wild-type (WT) mice as described (Brkic *et al*, 2015b). It has been shown that Aβ and its deposits cause chronic inflammation in AD, so this simple but reliable model is used to investigate how AβO interferes with neuronal processes and to test the efficacy of new therapeutic approaches (Akiyama *et al*, 2000; Balducci & Forloni, 2014). Six hours after icv injection of AβO, inflammation was evident in the choroid plexus as well as in the hippocampus. In both brain regions, multiple inflammatory and NF-κB-dependent genes were significantly upregulated. These include *Il6, Il1β, Nos2, Cxcl9,* and several MMPs, as well as *Nf-κb* itself (Fig 2C and D). Strikingly, the fold induction of the inflammatory genes in choroid plexus and hippocampus was hundred to thousand times higher in this model of AβO-induced toxicity than in the respective brain structures of APP/PS1$^{tg/wt}$ mice. Moreover, in contrast to the absence of *Tnf* expression in chronic AD pathology in choroid plexus of mouse and human, *Tnf* was significantly induced in the two brain structures in the acute AβO toxicity model. Also, the expression of *Tnfrsf1a* and *Tnfrsf1b* in the choroid plexus was also significantly higher.

### TNFR1 deficiency reduces the inflammatory signature in the choroid plexus and hippocampus in transgenic AD mice and upon AβO injection

Our observations in late-stage AD patients and in the two AD mouse models show that TNF and NF-κB signaling are important in AD pathology. As TNF signaling via TNFR1 is an important activator of NF-κB, we studied the effect of abrogating TNFR1 in the choroid plexus and hippocampus of AD mice. First, we crossed APP/PS1$^{tg/wt}$ mice with TNFR1-deficient mice and determined inflammatory gene expression in the choroid plexus and the hippocampus at late-stage AD mice and compared the results with age-matched control mice. We focused on NF-κB-dependent genes that were upregulated in one or both mouse models. Interestingly, though TNFR1 deficiency did not prevent the induction of all determined genes in the two brain regions of APP/PS1$^{tg/wt}$ mice, it did affect the expression of several potent inflammatory mediators. In the choroid plexus, expression of *Il6* and *Cxcl9* was significantly decreased, but *Nos2* expression was unaltered (Fig 3A–C). In agreement with the data in Fig 2A, *Nos2* and *Il1β* expression was not increased in the hippocampus of APP/PS1$^{tg/wt}$ mice, so it was also not affected by TNFR1 deficiency (Fig 3E and F). However, expression levels of *Cxcl9* and *Tnf* were significantly lower in APP/PS1$^{tg/wt}$ mice in the TNFR1$^{-/-}$ background compared to the WT background (Fig 3G and H). Interestingly, *Lcn2* expression was significantly induced in the choroid plexus and hippocampus of APP/PS1$^{tg/wt}$ mice (Fig 2A). In the choroid plexus, the induction was significantly weaker in the TNFR1$^{-/-}$ background (Fig 3D).

To investigate the effects of TNFR1 deficiency on the direct response to AβO, TNFR1$^{-/-}$ mice were injected icv with AβO or with scrambled peptide (control), and expression of inflammatory genes was studied in the choroid plexus and hippocampus 6 h later. Although AβO did significantly induce the expression of pro-inflammatory genes *Il6, Nos2,* and *Cxcl9* in the choroid plexus of TNFR1$^{-/-}$ mice compared to control mice, this induction was significantly reduced in the absence of TNFR1 compared to WT mice (Fig 3I and K). Also, the expression of *Il1β* and *Tnf* was significantly reduced in

this setting (data not shown). Strikingly, expression of *Il1β, Nos2, Cxcl9,* and *Tnf* in the hippocampus was completely abrogated in AβO-injected TNFR1$^{-/-}$ mice with similar levels as in scrambled injected controls (Fig 3M–P). In agreement with the results obtained in APP/PS1$^{tg/wt}$TNFR1$^{-/-}$ mice, also in the choroid plexus of AβO-injected TNFR1$^{-/-}$ mice *Lcn2* expression was reduced compared to WT mice (Fig 3L), but in the hippocampus there was no change (data not shown). In summary, our studies in both AD mouse models confirmed the importance of TNF/TNFR1 signaling in the inflammation associated with AD pathology.

### TNFR1 deficiency protects the choroid plexus against morphological changes induced by AβO and in APP/PS1 mice

The choroid plexus is morphologically altered in AD patients (Marques *et al*, 2013). We previously showed the direct detrimental effects of AβO on the morphology of CPE cells due to inflammation and impairment of the blood–CSF barrier (Brkic *et al*, 2015b). To examine the effect of TNFR1 deficiency on the morphology of the CPE cells, we injected WT and TNFR1$^{-/-}$ mice icv with either AβO or scrambled control and studied the structural alterations in CPE cells 6 h later by transmission electron microscopy (TEM) and volume scanning electron microscopy (SEM). Figure 4A and B shows that CPE cells of WT and TNFR1$^{-/-}$ mice injected with scrambled peptide appear morphologically normal. In contrast, CPE cells of AβO-injected WT mice lost their cuboidal structure and the microvilli on the apical side are fewer, shorter, and disorganized (Fig 4C). Conversely, the morphology of CPE cells of AβO-injected TNFR1$^{-/-}$ mice is preserved (Fig 4D): The cells are cuboidal and the microvilli appear normal. To better comprehend the overall differences in shape and ultrastructure of CPE cells between AβO-injected WT and TNFR1$^{-/-}$ mice, we made use of volume SEM (Brkic *et al*, 2015b). Serial block-face SEM (SBF-SEM) is an application used to generate electron microscopy image stacks that are reconstructed to generate 3D images (Denk & Horstmann, 2004). We performed SBF-SEM on *en bloc*-stained CP samples and the generated datasets were used for manual segmentation of the CPE, allowing the 3D visualization of the cell shape at nanometer resolution. As shown in Fig 5D, mainly the microvilli of the choroid plexus of AβO-injected TNFR1$^{-/-}$ mice were preserved compared to the choroid plexus of AβO-injected TNFR1$^{+/+}$ mice, whereas the typical cuboidal shape of the choroid plexus of TNFR1$^{-/-}$ also shows some alterations upon AβO injection (Fig 5A–D; Movies EV1–EV4).

To confirm these results in a transgenic model of AD, we dissected the choroid plexus from 18-week-old APP/PS1$^{tg/wt}$ mice in a WT and TNFR1$^{-/-}$ background and from age-matched non-transgenic counterparts. We investigated the structural alterations of the CPE cells with TEM (Fig EV1) and SEM (Fig EV2 and Movies EV5–EV8) and found that 18-week-old non-transgenic littermates, that is, APP/PS1$^{wt/wt}$ mice both in WT and TNFR1$^{-/-}$ background, exhibited almost no morphological changes (Fig EV1A and B). Some of the CPE cells show a limited alteration in their cuboidal cell structure as they are more point-shaped; however, this can be considered as a normal process in the choroid plexus (Gorle *et al*, 2016). The cytoplasm and mitochondria are nicely preserved in mice of that age. Contrastingly, TEM images of the CPE cells of age-matched APP/PS1$^{tg/wt}$ mice in a WT background displayed a profound

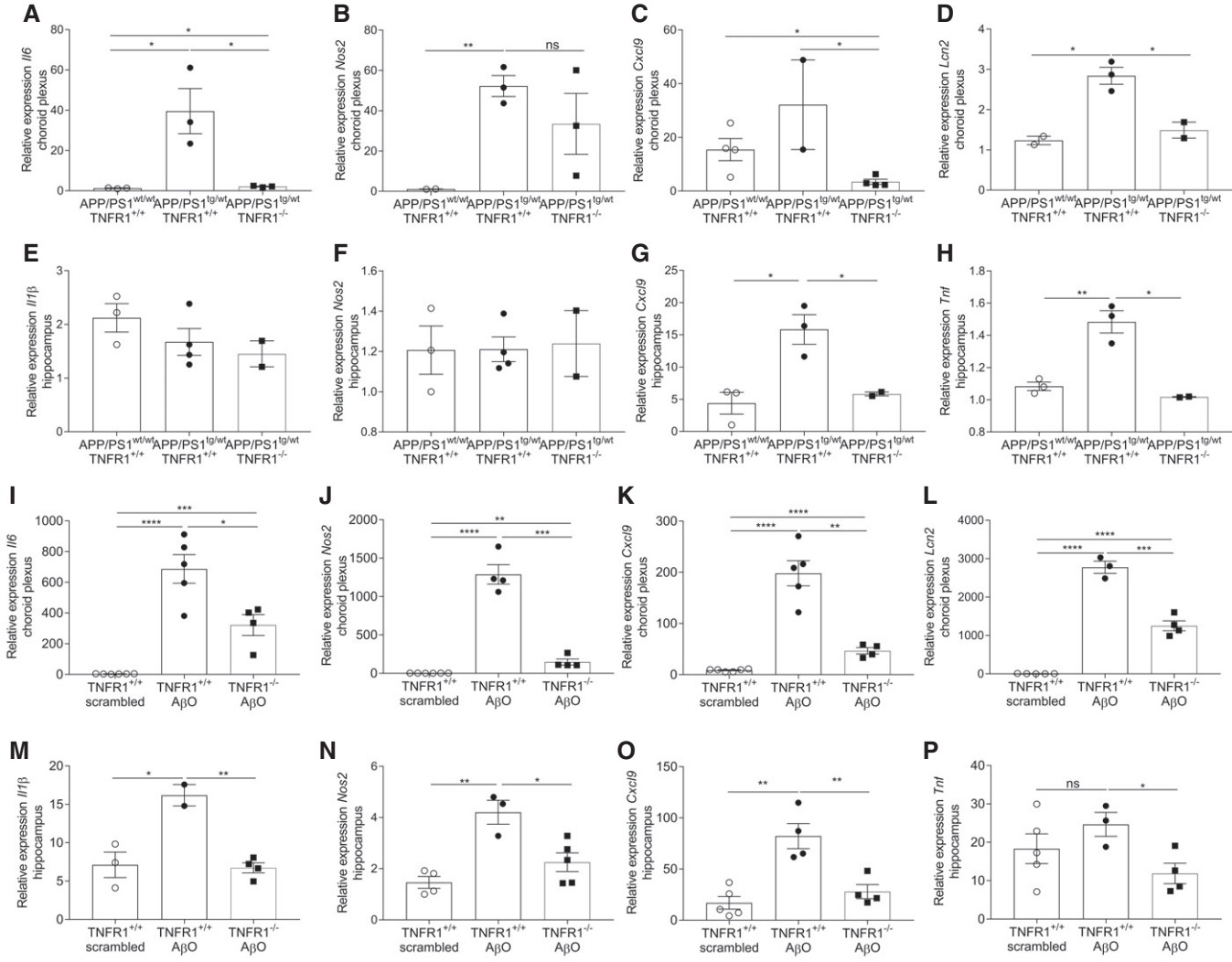

**Figure 3.  TNFR1 deficiency abrogates inflammation in choroid plexus and hippocampus.**

A–H   Relative mRNA gene expression of *Il6*, *Nos2*, *Cxcl9*, and *Lcn2* in the choroid plexus (A–D) and of *Il1β*, *Nos2*, *Cxcl9*, and *Tnf* in the hippocampus (E–H) of late-stage AD APP/PS1^tg/wt mice in a TNFR1^+/+ and TNFR1^−/− background compared to age-matched APP/PS1^wt/wt mice (n = 2–4/group).

I–P   Relative mRNA gene expression of *Il6*, *Nos2*, *Cxcl9*, and *Lcn2* in the choroid plexus (I–L) and of *Il1β*, *Nos2*, *Cxcl9*, and *Tnf* in the hippocampus (M–P) of C57BL/6J TNFR1^+/+ and TNFR1^−/− mice 6 h after intracerebroventricular (icv) injection with scrambled peptide or with AβO (1 μg/ml) (n = 4–5/group).

Data information: Bars represent mean ± SEM. qPCR was normalized to stable housekeeping genes, determined by GeNorm. Statistics were performed with an unpaired *t*-test, *0.01 ≤ P < 0.05; **0.001 ≤ P < 0.01; ***0.001 ≤ P < 0.0001, ****P < 0.0001.

transformation into cells with a more degenerative state (Fig EV1C). Indeed, the cytoplasm of these cells is translucent, indicative for cellular degradation and the nuclei have irregular shapes (Fig EV1C, zoom). It is also clear that the capillaries of the choroid plexus are swollen and dilated, and filled with a lot of red blood cells. TNFR1 deficiency in APP/PS1^tg/wt mice clearly protects the morphology of the CPE cells at several levels. There are less signs of cellular degradation, the CPE cells are still cuboidal as observed in non-transgenic littermates, and the mitochondria and nuclear shape are preserved (Fig EV1D and zoom). Also the capillaries are less swollen and less red blood cells are observed. In contrast to what we observed in AβO-injected mice, the transgenic APP/PS1^tg/wt mice showed no severe loss of microvilli alignment (Figs 4C and D, and EV1C and

D). Next, SEM analysis of the CPE cells of APP/PS1 mice confirmed our observations made by the TEM: CPE cells of APP/PS1^wt/wt mice show some signs of aging-related damage which is much more pronounced in a APP/PS1^tg/wt mice and this is prevented in TNFR1^−/− background (Fig EV2A–D, Movies EV5–EV8).

**TNFR1 deficiency protects against blood–CSF barrier impairment by reduced MMP expression and maintenance**

To investigate the mechanisms at the molecular level behind the protection in TNFR1^−/− mice, we examined blood–CSF barrier permeability by measuring leakage of 4 kDa FITC-dextran into the CSF 6 h after icv injection of AβO. We previously reported that icv

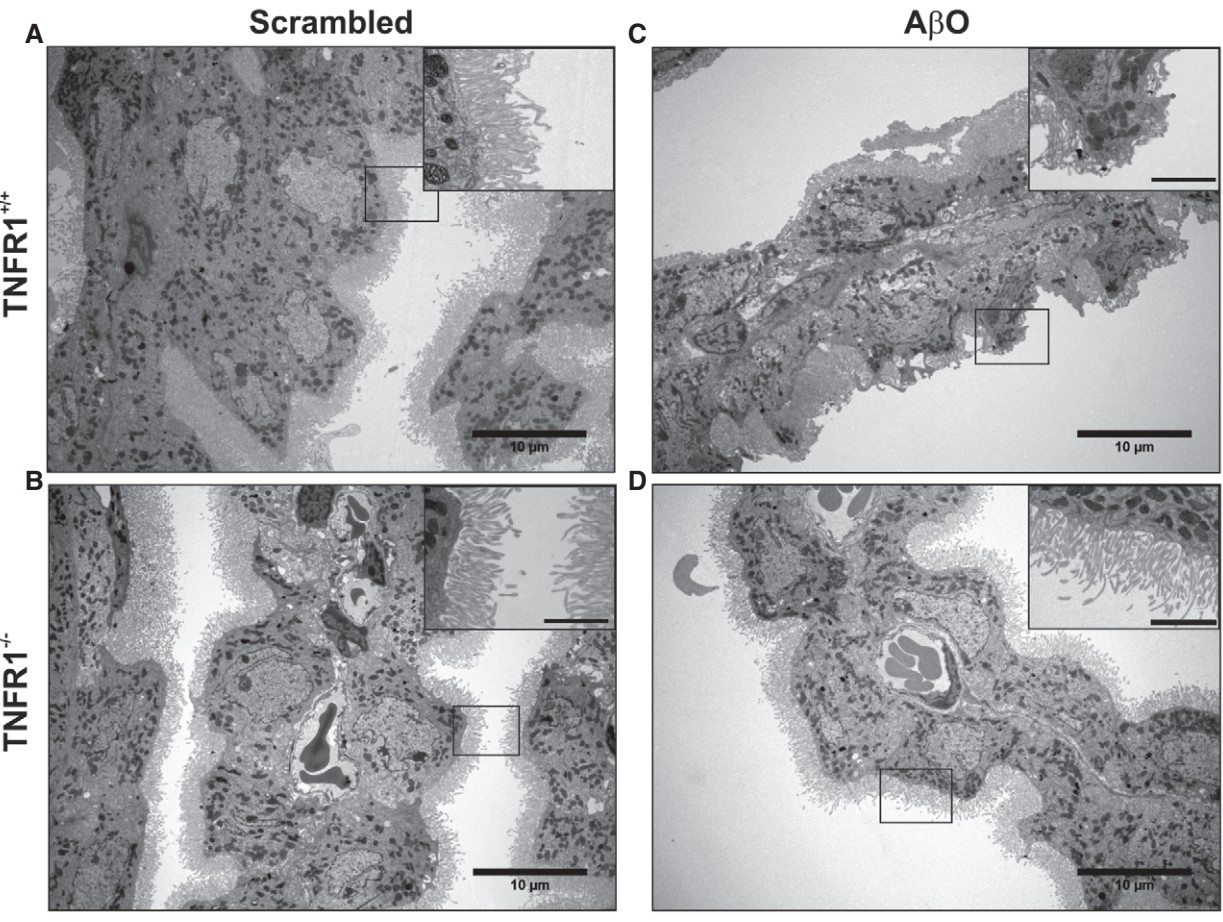

**Figure 4.  TNFR1 deficiency protects against AβO-induced morphological alterations of the choroid plexus determined by TEM.**

A–D   Representative conventional transmission electron microscopy (TEM) images of the choroid plexus 6 h after intracerebroventricular (icv) injection of scrambled peptide (control) (A, B) or Aβ$_{1-42}$ oligomers (AβO, 1 μg/ml) (C, D) in C57BL/6J TNFR1$^{+/+}$ (A, C) and TNFR1$^{-/-}$ (B, D) mice (*n* = 2/group). In scrambled-injected control mice (A, B), the cuboidal structure of the choroid plexus epithelial (CPE) cells is preserved and the microvilli are aligned and structured (*insert*). The cuboidal structure of the CPE cells in AβO-injected WT mice (C) is altered, and the microvilli are shortened (*insert*). In contrast (D), CPE cells of TNFR1$^{-/-}$ mice icv injected with AβO have a normal morphology and the microvilli are organized (*insert*). The TEM images were taken at a magnification of 1,000×, and scale bar represents 10 μm; inserts were taken at a magnification of 3,000×, and scale bar represents 2 μm.

injection of AβO affects blood–CSF barrier integrity and disturbs barrier function (Brkic *et al*, 2015b). Here, AβO significantly increased blood–CSF permeability in WT mice compared to control mice injected with scrambled peptide (Fig 6A). In contrast, icv AβO injection did not induce blood–CSF barrier leakage in TNFR1$^{-/-}$ mice. Given that disruption of the blood–CSF barrier has been linked to increased activity of matrix metalloproteinases (MMPs) in the choroid plexus (Brkic *et al*, 2015b), we investigated their expression 6 h after icv injection of AβO. Expression of both *Mmp3* and *Mmp8* in the choroid plexus was significantly increased after icv AβO injection to what we reported previously (Brkic *et al*, 2015b), but this increase was significantly less in TNFR1-deficient mice (Fig 6B and C).

As the integrity of the blood–CSF barrier is maintained by junctional proteins between the CPE cells, and MMPs are known to disrupt tight junctions (TJs) (Vandenbroucke *et al*, 2012; Brkic *et al*, 2015b), we used qPCR and immunohistochemistry to determine whether TJs in the choroid plexus of TNFR1$^{-/-}$ mice were preserved after AβO injection. In line with previous observations in

mouse models and human AD patients (Bergen *et al*, 2015; Brkic *et al*, 2015b), 6 h after AβO injection claudin-5 (*Cldn5*) expression was significantly downregulated in WT mice but not in TNFR1$^{-/-}$ mice (Fig 6D). The expression of occludin (*Ocln*) after AβO injection was also significantly weaker in WT mice but not in TNFR1$^{-/-}$ mice (Fig 6E). Claudin-1 (*Cldn1*) mRNA levels were unaltered upon AβO injection in both genotypes (data not shown), but the subcellular localization of CLDN1 does show alterations (Fig 6F). Six hours after AβO injection, CLDN1 remained at the apical side of the CPE cells of injected TNFR1$^{-/-}$ mice (Fig 6F, arrows), whereas in WT mice it became more diffuse, its expression weakened, and is less obviously prominent in the apical region (Fig 6F, arrowheads).

**TNFR1 deficiency reduces Aβ levels and plaque disposition in APP/PS1 mice**

We further assessed the pathological consequences of TNF/TNFR1 signaling in the transgenic AD mice by characterizing different

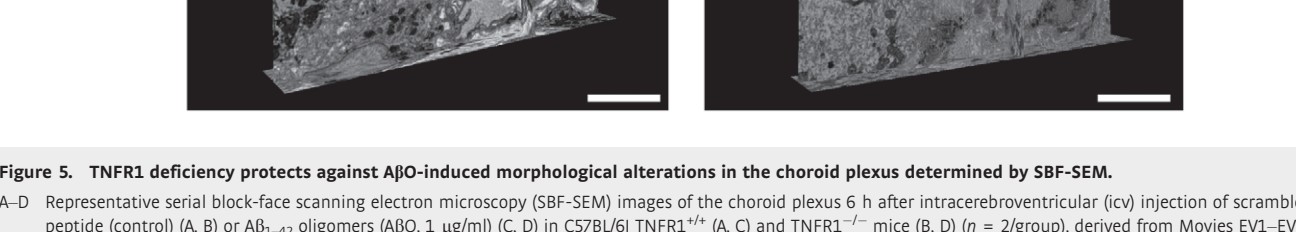

**Figure 5. TNFR1 deficiency protects against AβO-induced morphological alterations in the choroid plexus determined by SBF-SEM.**

A–D  Representative serial block-face scanning electron microscopy (SBF-SEM) images of the choroid plexus 6 h after intracerebroventricular (icv) injection of scrambled peptide (control) (A, B) or Aβ$_{1-42}$ oligomers (AβO, 1 μg/ml) (C, D) in C57BL/6J TNFR1$^{+/+}$ (A, C) and TNFR1$^{-/-}$ mice (B, D) (*n* = 2/group), derived from Movies EV1–EV4. CSF: cerebrospinal fluid; Mv, microvilli; Nu, nucleus. Scale bar, 5 μm.

hallmarks of AD. APP/PS1 mice aged 4–12 months display an increase in plaque burden (Radde *et al*, 2006). To determine whether genetic inactivation of TNFR1 influences Aβ plaque formation in APP/PS1$^{tg/wt}$ mice, we visualized the Aβ plaques with Thioflavin-S in brain sections of APP/PS1$^{tg/wt}$ mice in a WT and a TNFR1$^{-/-}$ background. Thioflavin-S stains mature Aβ plaques in the β-sheet conformation and the signal increases with increased Aβ oligomerization. In contrast to age-matched control mice (not shown), APP/PS1$^{tg/wt}$ mice showed plaque formation (Fig 7A, C, and D), and the brains of TNFR1-deficient APP/PS1$^{tg/wt}$ mice had significantly less Thioflavin-S$^+$ areas. This indicates that β-sheet formation and thus probably Aβ oligomerization are reduced in the absence of TNFR1, although this reduction is fairly modest. As the size of the plaques is indicative of the severity of the Aβ pathology (Zhou *et al*, 2005), we performed a morphometric analysis on the

brain sections stained with Thioflavin-S (Fig 7B). This analysis demonstrates that there are significantly fewer of the smaller plaques (< 10 μm²) in APP/PS1$^{tg/wt}$ mice deficient in TNFR1 and that there is a tendency for medium-sized Aβ plaques (10–20 μm²) to also be fewer. However, no reduction in number was observed in larger Aβ plaques (> 20 μm²).

Since the genetic deletion of TNFR1 reduces Aβ burden in transgenic AD mice, we questioned whether TNFR1 deficiency in APP/PS1$^{tg/wt}$ mice can affect total Aβ levels. There is strong evidence that mainly soluble Aβ drives disease progression in AD rather than the amyloid deposits themselves, and Aβ$_{42}$ is considered the most neurotoxic form (Chen & Glabe, 2006). Therefore, we used ELISA to measure the levels of both soluble and insoluble Aβ$_{40}$ (S-Aβ$_{40}$ and IS-Aβ$_{40}$) and Aβ$_{42}$ (S-Aβ$_{42}$ and IS-Aβ$_{42}$) in the cortex of APP/PS1$^{tg/wt}$ mice, because the amyloid plaque deposition starts at the age of

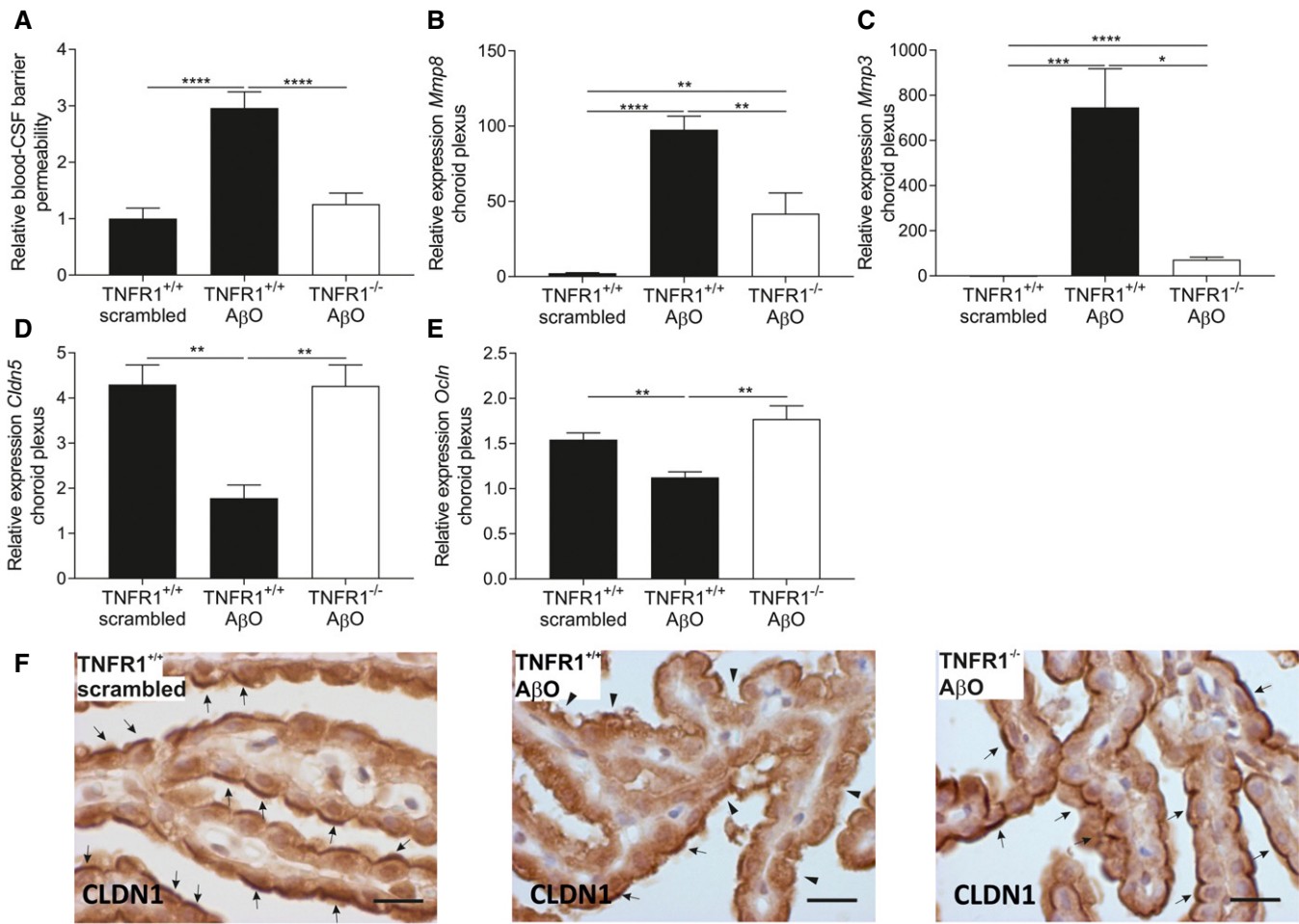

**Figure 6.   TNFR1 deficiency protects the blood–cerebrospinal fluid (CSF) barrier by preventing MMP increase and preserving tight junctions.**

A    Blood–CSF barrier permeability was determined by measuring leakage of FITC-labeled dextran in the CSF of C57BL/6J TNFR1$^{+/+}$ and TNFR1$^{-/-}$ mice 6 h after intracerebroventricular (icv) injection of either scrambled peptide or A$\beta_{1-42}$ oligomer (A$\beta$O 1 µg/ml) ($n$ = 11–14/group).

B–E   Relative mRNA gene expression of *Mmp8, Mmp3, Cldn5,* and *Ocln* in choroid plexus of TNFR1$^{+/+}$ and TNFR1$^{-/-}$ mice 6 h after icv injection of 1 µg/ml A$\beta$O ($n$ = 5–6/group).

F    Representative images of CLDN1 staining in choroid plexus of the fourth ventricle of TNFR1$^{+/+}$ (*left and middle image*) and TNFR1$^{-/-}$ mice (*right image*) 6 h after icv injection with either 1 µg/ml A$\beta$O (*middle and right image*) or scrambled peptide (*left image*). Arrows indicate preserved CLDN1 tight junctions, and arrowheads indicate affected CLDN1 tight junctions ($n$ = 3/group). Scale bar represents 15 µm.

Data information: Bars represent mean ± SEM. qPCR was normalized to stable housekeeping genes determined by GeNorm. The experiments are done in duplicates, and pooled results are shown in (A) and representative results in (B–E). Statistics were performed with a one-way ANOVA for the permeability data (A) and an unpaired *t*-test for the qPCRs (B–E), *$0.01 \leq P < 0.05$; **$0.001 \leq P < 0.01$; ***$0.001 \leq P < 0.0001$, ****$P < 0.0001$.

about 6 weeks in this region (Maia *et al*, 2013). As shown in Fig 7E– H, the levels of all A$\beta$ peptides were significantly higher in the corti- cal brain fraction of APP/PS1$^{tg/wt}$ mice than in non-transgenic litter- mates of the same age. In APP/PS1$^{tg/wt}$ TNFR1$^{-/-}$ mice, S-A$\beta_{42}$ levels were significantly lower (Fig 7F). Moreover, in line with the A$\beta$ plaque deposition, the cortical levels of IS-A$\beta_{42}$ were also decreased, although not significantly, compared to WT APP/PS1$^{tg/wt}$ mice (Fig 7H). Conversely, the levels of S-A$\beta_{40}$ and IS-A$\beta_{40}$ were not significantly changed in the absence of TNFR1 (Fig 7E and G).

Previous studies reported that TNFR1 signaling affects the produc- tion of A$\beta$ by inducing the transcription of beta-secretase 1 (BACE1) (He *et al*, 2007; Yamamoto *et al*, 2007). As we could show that dele- tion of TNFR1 in APP/PS1$^{tg/wt}$ mice results in a reduced A$\beta$

pathology, we determined the gene expression levels of *Bace1* in the hippocampus of APP/PS1$^{tg/wt}$ in a WT and TNFR1$^{-/-}$ background and compared it with age-matched non-transgenic littermates. As expected, *Bace1* expression was significantly increased in APP/PS1$^{tg/wt}$ mice, but importantly, this induction was completely prevented when the TNF/TNFR1 signaling pathway is deleted (Fig 7I).

**TNFR1 deficiency prevents microglia activation in APP/PS1 mice and upon icv injection of A$\beta$O**

Microglia activation is a prominent hallmark of AD pathology, and it is well documented that both A$\beta$O and A$\beta$ fibrils activate microglia (Sondag *et al*, 2009). Activated microglia propagate brain

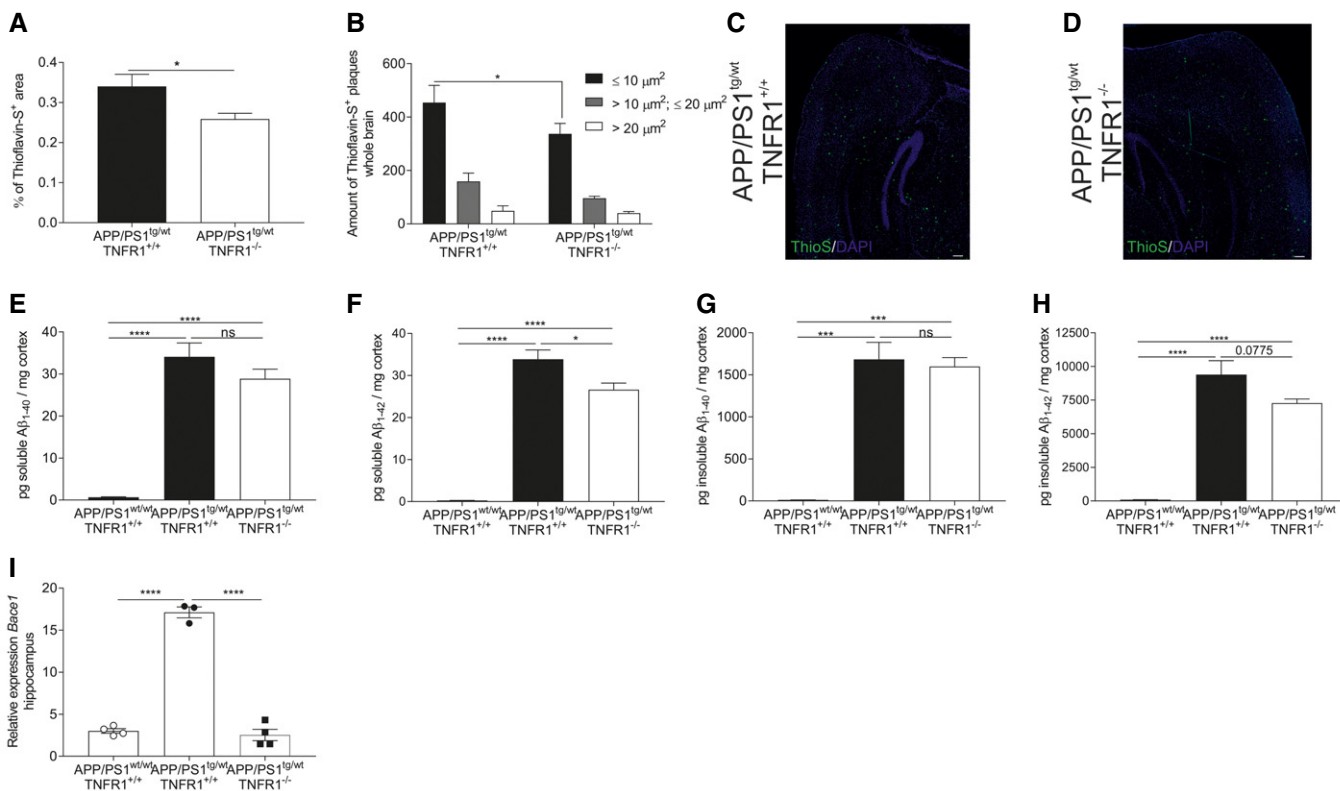

**Figure 7. TNFR1 deficiency reduces Aβ pathology in APP/PS1 mice.**

A–D   Brain sections of late-stage C57BL/6J APP/PS1$^{tg/wt}$ and APP/PS1$^{tg/wt}$TNFR1$^{-/-}$ mice were stained with Thioflavin-S to detect Aβ disposition in the whole brain. The amount of plaques was quantified (A), and a morphometric analysis was performed (B) (*n* = 5 and 6 mice/group, respectively). Representative images (C, D) of Thioflavin-S (ThioS) staining of the brain containing the hippocampus of late-stage APP/PS1$^{tg/wt}$ and age-matched APP/PS1$^{tg/wt}$TNFR1$^{-/-}$ mice (scale bar represents 200 μm).

E–H   ELISA analysis of soluble and insoluble Aβ$_{1-40}$ and Aβ$_{1-42}$ in the cortex of late-stage APP/PS1$^{tg/wt}$ in a TNFR1$^{+/+}$ and TNFR1$^{-/-}$ background compared to age-matched controls (*n* = 8, 9, and 3 mice, respectively).

I   Relative mRNA expression of *Bace1* in the hippocampus of late-stage APP/PS1$^{tg/wt}$ mice in a TNFR1$^{+/+}$ and TNFR1$^{-/-}$ background compared to age-matched APP/PS1$^{wt/wt}$ mice (*n* = 5/group).

Data information: Bars represent mean ± SEM. qPCR was normalized to stable housekeeping genes determined by GeNorm. Statistics were performed with an unpaired *t*-test (A), a two-way ANOVA assay (B) or a one-way ANOVA (E–I), *0.01 ≤ *P* < 0.05; ***0.001 ≤ *P* < 0.0001; ****P* < 0.0001.

---

inflammation further and induce neuronal cell loss by releasing neurotoxic mediators in the brain (Parajuli *et al*, 2013). As this process plays a pivotal role in the progression of AD, we assessed microglial activation in the two different AD models by examining brain sections stained for IBA1, which is a microglia-specific marker. Given the inflammatory signature in choroid plexus and hippocampus of APP/PS1$^{tg/wt}$ and AβO-injected WT mice, and the reduction in TNFR1$^{-/-}$ mice, we quantified the microglia on whole-brain sections. Counting IBA1-positive microglia showed an increase in brains of late-stage APP/PS1$^{tg/wt}$ mice (Fig 8A and B) and in brains of WT mice 6 h after icv injection of AβO (Fig 8C and D), compared to their respective controls. In both AD models, this was significantly reduced in TNFR1-deficient background (Fig 8A–D).

## TNFR1 deficiency protects against memory decline in APP/PS1$^{tg/wt}$ mice and induced by icv injection of AβO

Cognitive impairment is one of the most important hallmarks of AD (Ledo *et al*, 2013; Ferreira *et al*, 2015; Jongbloed *et al*, 2015).

Cognitive decline (spatial learning, measured by the radial four-arm maze assay) has been reported in APP/PS1 mice (Radde *et al*, 2006). Microglia activation precedes cognitive decline in several AD mouse models (Parajuli *et al*, 2013), and soluble AβO is involved in the induction of cognitive impairment (Ferreira *et al*, 2015).

In our study, we evaluated cognitive behavior using the novel object recognition test (NOR), which assesses working memory based on the differences in exploration time of novel and familiar objects (Antunes & Biala, 2012). We found that both short-term memory (STM) and long-term memory (LTM) in APP/PS1$^{tg/wt}$ mice were indeed diminished compared to age-matched control mice: Their NOR performance was significantly lower: 65% vs. 49%, *P* < 0.001 for STM, and 64% vs. 51%, *P* = 0.02 for LTM. Though for STM the novel object preference of TNFR1-deficient APP/PS1$^{tg/wt}$ mice was not significantly better (57%) than that of APP/PS1$^{tg/wt}$ mice with functional TNFR1 (49%) (Fig 9A), for LTM their novel object preference was significantly better (67% vs. 51%, *P* = 0.01) (Fig 9B). Moreover, LTM assessment revealed that the preference of APP/PS1$^{tg/wt}$ TNFR1$^{-/-}$ mice for novel

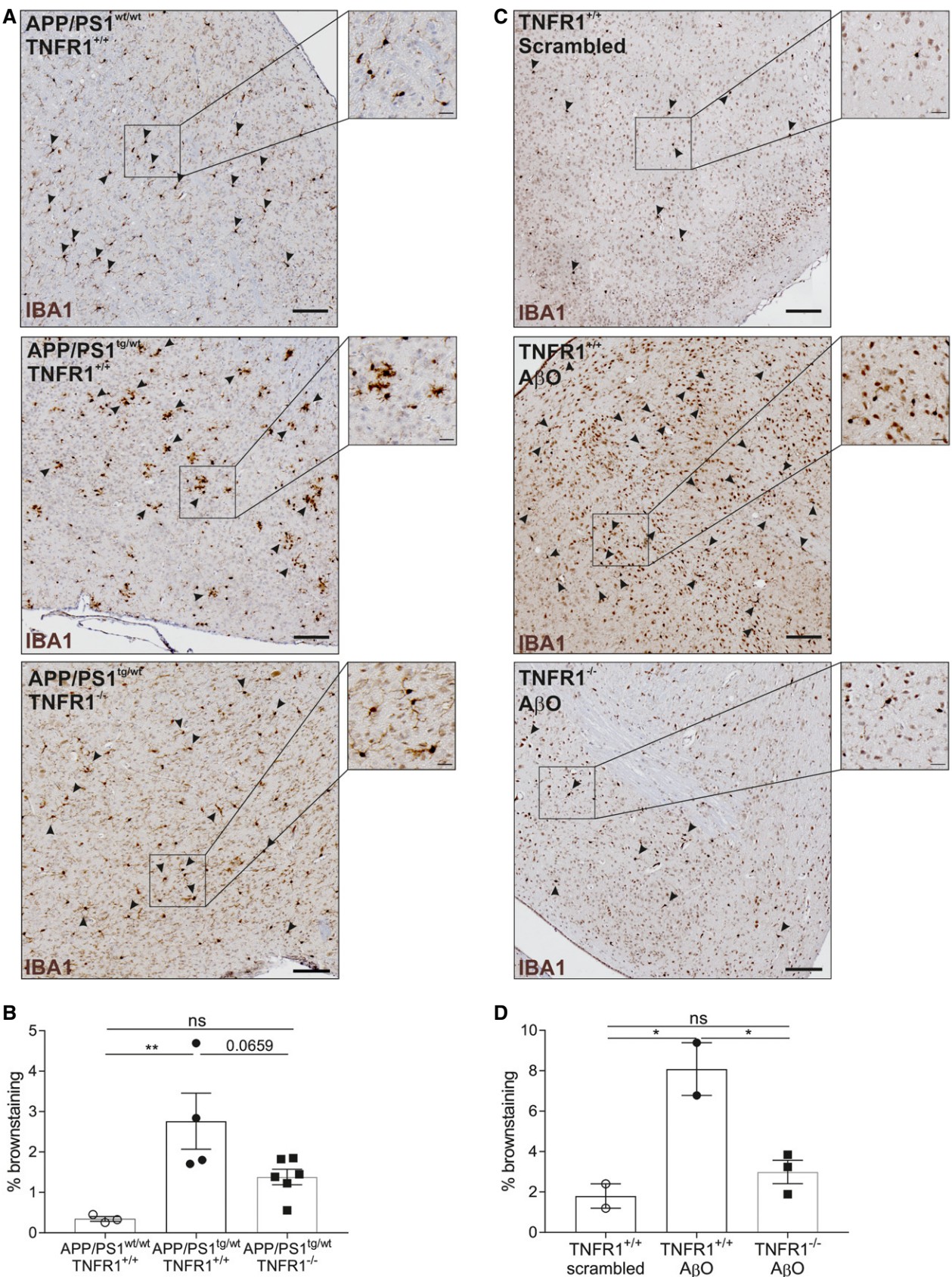

**Figure 8.**

◄

**Figure 8. TNFR1 deficiency prevents microglia activation in APP/PS1 mice and upon icv AβO injection.**

A, B  IBA1 staining for microglia on whole-brain sections of late-stage C57BL/6J APP/PS1$^{tg/wt}$ in a TNFR1$^{+/+}$ and TNFR1$^{-/-}$ background compared to age-matched controls. (A) Representative images of a region around the fourth ventricle (microglia indicated with arrowheads) in age-matched controls (*upper panel, n* = 3), APP/PS1$^{tg/wt}$TNFR1$^{+/+}$ (*middle panel, n* = 4), and APP/PS1$^{tg/wt}$TNFR1$^{-/-}$ (*lower panel, n* = 6) mice. (B) Quantification of IBA1$^+$ cell count (determined by brown staining).

C, D  IBA1 staining for microglia on whole-brain sections of C57BL/6J TNFR1$^{+/+}$ mice 6 h after intracerebroventricular (icv) injection with either scrambled peptide or Aβ$_{1-42}$ oligomer (AβO, 1 μg/ml) or TNFR1$^{-/-}$ mice icv injected with AβO. (C) Representative images of a region around the fourth ventricle (microglia indicated with arrowheads) in scrambled-injected mice (*upper panel, n* = 2) and in AβO-injected TNFR1$^{+/+}$ (*middle panel, n* = 3) and TNFR1$^{-/-}$ mice (*lower panel, n* = 3). (D) Quantification of IBA1$^+$ cell count (determined by brown staining).

Data information: Scale bars represent 100 μm and insert 20 μm. Bars represent mean ± SEM. Statistics were performed with a one-way ANOVA assay, *$0.01 \leq P < 0.05$; **$0.001 \leq P < 0.01$.

objects was as good as that of the non-transgenic WT mice of the same age (64% vs. 67%).

Next, we assessed STM and LTM with the NOR test in WT mice upon icv injection of AβO and compared this with WT mice injected with scrambled peptide and TNFR1$^{-/-}$ mice injected with AβO. As expected, WT mice injected with scrambled peptide had a novel object preference of about 67%, indicating a functional STM. In contrast, WT mice injected with AβO showed a reduced novel object preference (53%), indicative for AβO-induced memory decline ($P = 0.003$). However, in TNFR1$^{-/-}$ mice, icv injection of AβO did not lead to STM decline, and their novel object preference (63%) was significantly higher than in AβO-injected WT mice ($P = 0.046$; Fig 9C). The same pattern was observed for LTM (Fig 9D). AβO injection in TNFR1$^{-/-}$ mice resulted in LTM performance resembling that of mice injected with scrambled peptide (60% vs. 62%, respectively) and was significantly better than AβO-injected WT mice (53%, $P = 0.04$). Collectively, these results demonstrate that TNFR1 deficiency interferes with the AβO-induced brain inflammation and ameliorates the subsequent memory impairment. This finding indicates that TNFR1 has an important role in AD pathogenesis and its associated cognitive decline.

**Treatment with an anti-TNFR1 biologic recapitulates the effects of TNFR1 deficiency**

Our results show that TNF/TNFR1 signaling is involved in the pathology of AD, which makes it a potential therapeutic target. We wanted to provide full proof of concept for therapeutic TNFR1-targeting by assessing both STM and LTM. For this experiment, we used an in-house generated trivalent anti-TNFR1 Nanobody called TROS (Steeland *et al*, 2015). Nanobodies have many advantages compared to antibodies when they are intended for chronic therapy or brain delivery (Steeland *et al*, 2016). Since TROS inhibits human TNFR1 (hTNFR1) but shows no cross-reactivity with mouse TNFR1, therapeutic studies with TROS have to be performed in transgenic mice that express hTNFR1 in a mouse TNFR1$^{-/-}$ background (hTNFR1 Tg) (Steeland *et al*, 2017). First, to exclude any TROS or Nanobody-related effect on inflammation upon icv injection, we administered TROS together with AβO via icv injection in WT mice which lack the therapeutic target of TROS, followed by choroid plexus isolation and qPCR analysis. We determined the expression of several inflammatory genes, for example, *Il6, Il1β, Tnf, Mmp3,* and *Tnfrsf1a* in the choroid plexus upon AβO injection together with PBS and compared it with AβO/TROS co-injection. These results show that TROS neither induces nor blocks AβO-induced inflammation in the absence of its target (Fig EV3). Next, we assessed the effect of TROS on the memory performance in AβO-injected hTNFR1

Tg mice, which are mice that express the target of TROS. The preference for novel objects of hTNFR1 Tg mice injected with scrambled peptide resembles that of normal WT mice (65%), indicating that their STM is sustained at normal levels. When these mice were icv injected with AβO, the preference for novel objects declined due to STM impairment (53%, $P = 0.03$). Co-injection of TROS (1.55 μg/μl) with AβO (1 μg/ml) significantly prevented STM decline: Preference for novel objects was the same as in hTNFR1 Tg mice injected with scrambled peptide (68%, $P = 0.007$; Fig 9E). Similar results were obtained for LTM: Preference for novel objects of mice co-injected with TROS was not significantly worse than the scrambled peptide mouse group (65% vs. 79%, $P = 0.052$) but not significantly better than in mice co-injected with vehicle (56%, $P = 0.2$; Fig 9F). In conclusion, our data demonstrate that TNFR1 is a valuable new drug target to treat AβO-induced cognitive decline.

## Discussion

Alzheimer's disease (AD) is a chronic, progressive, neurodegenerative disease with a totally unmet need for drugs that prevent the continuing and long-term cognitive decline, as no FDA-approved drug meets this criterion (Citron, 2010). Neuroinflammation, a prominent hallmark of AD, occurs very early during disease progression and is mediated by the production of pro-inflammatory mediators and activated glial cells that affect different brain regions (Akiyama *et al*, 2000). Epidemiological studies linked the use of anti-inflammatory medication to a lower incidence of AD, but strategies to target the relevant pathways in patients with established AD have not been successful (Miguel-Alvarez *et al*, 2015). TNF is a very potent pleiotropic pro-inflammatory cytokine, and there is a vast body of data showing its involvement in the pathology of AD. In the brain, TNF is expressed by microglia, astrocytes, and neurons, and genetic polymorphisms associated with TNF upregulation have been linked to AD susceptibility (Ramos *et al*, 2006). TNF signals via TNFR1 and TNFR2. In agreement with the harmful role of TNFR1 in other neurodegenerative diseases (Probert, 2015), several studies in mice have shown that this signaling pathway is detrimental in AD (Li *et al*, 2004; He *et al*, 2007). The role of TNFR1 contrasts sharply with that of TNFR2, to which multiple beneficial functions in the CNS have been attributed, such as its prominent role in protection of neurons against Aβ toxicity (Shen *et al*, 1997; Jiang *et al*, 2014; Probert, 2015).

In our study, we demonstrated that in late-stage AD patients, *TNF* as well as *IL1B* and *IL6* are the main upstream regulatory cytokines in the choroid plexus resulting in TNFR1-dependent activation of NF-κB. This potent transcription factor, which is activated

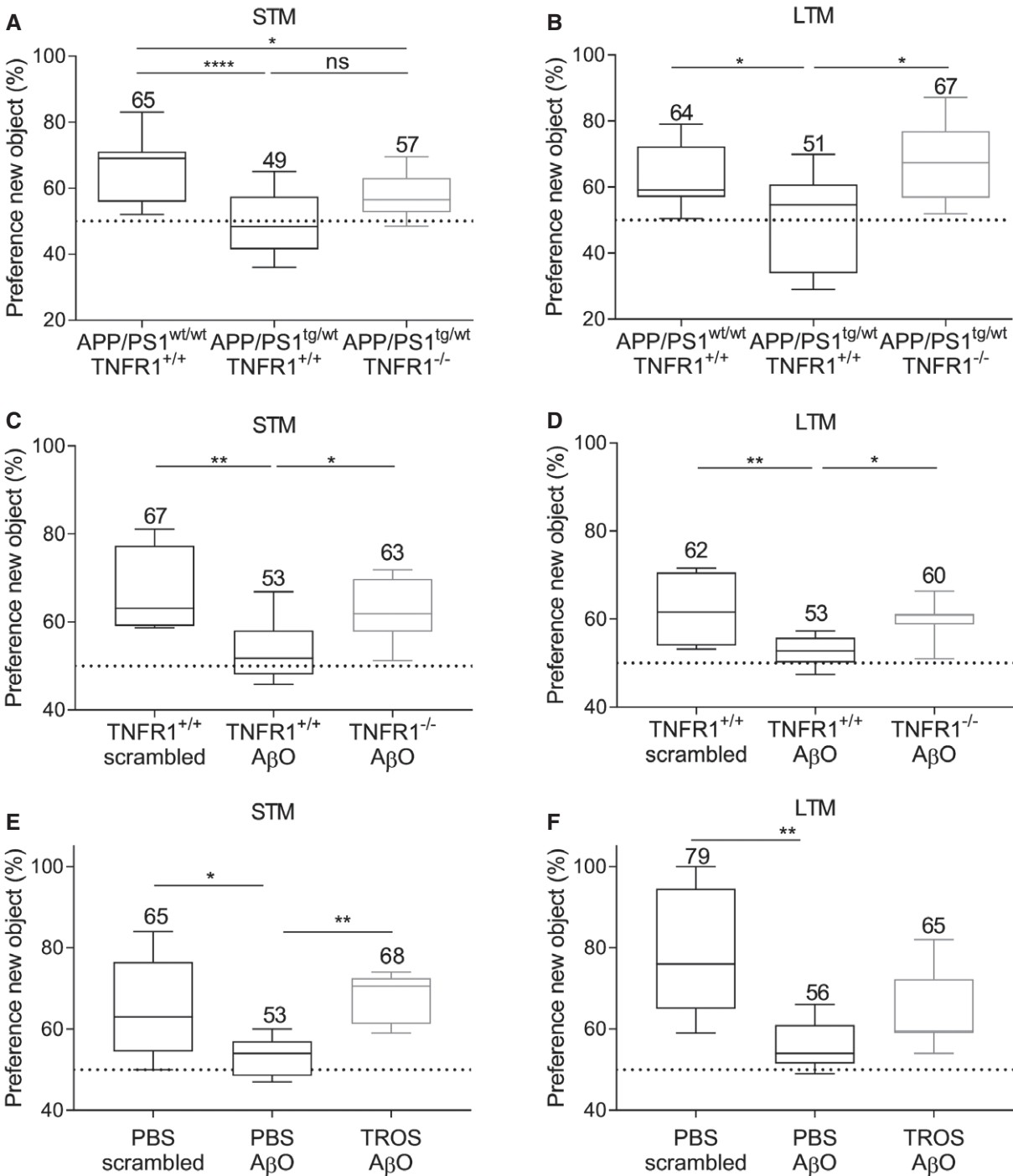

**Figure 9.  Genetic or therapeutic blockage of TNFR1 prevents cognitive decline in APP/PS1$^{tg/wt}$ mice and upon icv AβO injection.**

Analysis of short-term memory (STM) and long-term memory (LTM) evaluated by the novel object recognition (NOR) test.

A, B    STM (A) and LTM (B) assessed in C57BL/6J APP/PS1$^{wt/wt}$TNFR1$^{+/+}$ mice ($n = 19$) and APP/PS1$^{tg/wt}$ mice in a TNFR1$^{+/+}$ ($n = 10$) and TNFR1$^{-/-}$ background ($n = 11$) aged 30–34 weeks.

C, D    STM (C) and LTM (D) assessed 24 h after intracerebroventricular (icv) injection of either scrambled peptide or Aβ$_{1-42}$ oligomer (AβO, 1 μg/ml) in C57BL/6J TNFR1$^{+/+}$ or TNFR1$^{-/-}$ mice ($n = 6–11$/group).

E, F    STM (E) and LTM (F) assessed in human TNFR1 transgenic (hTNFR1 Tg) mice 24 h after icv injection of scrambled peptide, AβO alone (1 μg/ml), or AβO combined with a trivalent anti-TNFR1 Nanobody (TROS, 1.55 μg/μl) ($n = 6–8$/group).

Data information: Min-to-max box plots represent median with the 25 and 75% percentiles, and whiskers are the min and max values ± SEM. The mean values are indicated above the graphs. All experiments were done in duplicates, and pooled results are shown. Statistics were performed with a one-way ANOVA assay, *0.01 ≤ P < 0.05; **0.001 ≤ P < 0.01; ****P < 0.0001.

by TNF/TNFR1 interaction, elevates the expression of several inflammatory genes and amplifies neuroinflammation (Yang et al, 2002a; Wajant & Scheurich, 2011). The expression of *TNF* itself is not increased in late-stage AD patients, indicating that TNF is an early inflammatory inducer in the AD pathology that initiates the inflammatory cascade in the choroid plexus. A limitation of our study is the small age difference between the healthy subjects and AD patients. Consequently, some of the changes might be caused by aging, rather than Alzheimer's disease. Therefore, we also studied the involvement of NF-κB in two different mouse models of AD. As a first model of amyloidosis, we used transgenic mice expressing human amyloid precursor protein (APP) and presenilin 1 (PS1). This transgenic AD model displays several cellular and behavioral characteristics of AD, including development of cerebral Aβ plaques and cognitive impairment (Radde et al, 2006). For the second model, we injected oligomerized $A\beta_{1-42}$ (AβO) into the brain ventricles to study the direct toxic effects of AβO species. Also this model mimics key pathological characteristics of AD, such as tau hyperphosphorylation, neuronal cell loss, and impairment of hippocampus-dependent memory, making this a relevant model to study the amyloid-driven component of AD pathogenesis (Nitta et al, 1997; Brouillette et al, 2012; Cetin et al, 2013; Balducci & Forloni, 2014).

In both models, we confirmed NF-κB-induced inflammation in the choroid plexus and hippocampus. Similar to the choroid plexus expression in late-stage AD patients, the expression of *Tnf* was not elevated in the choroid plexus of late-stage APP/PS1 mice. Conversely, *Tnf* is highly expressed in both choroid plexus and hippocampus of wild-type (WT) mice icv injected with AβO. Also the expression of other inflammatory genes was 100–1,000 times higher after AβO injection in WT mice compared to the respective expression in APP/PS1 mice. This might be partially explained by the fact that the injected AβO is fourfold higher than the levels measured in APP/PS1 mice (Maia et al, 2013). Moreover, chronic exposure to an inflammatory trigger such as Aβ might exhaust and/or desensitize inflammation-responsive cells. As APP/PS1 mice are continuously exposed to toxic Aβ species throughout their life, this might explain the lower induction of inflammatory genes in these mice. Interestingly, also the expression of lipocalin 2 (*Lcn2*), which is known to silence TNFR2-mediated neuroprotective signaling cascades, is elevated in the choroid plexus of mice in both models (Naude et al, 2012).

One report showed that $A\beta_{1-40}$ physically interacts with TNFR1, thereby contributing to neuronal death (Li et al, 2004), and here we demonstrate that TNFR1 deficiency prevents AβO-induced inflammation in the choroid plexus and hippocampus. Based on these findings, we suggest that not only TNF but also AβO, potentially via direct TNFR1 binding, is responsible for the inflammation in these two brain regions.

The choroid plexus is an often neglected brain structure that contains one of the CNS barriers, called the blood–cerebrospinal fluid (CSF) barrier. Several studies have shown structural alterations in the morphology and functionality of the choroid plexus in AD patients, including epithelial atrophy, irregular basement thickening of CPE cells, and decreased CSF secretion (Serot et al, 2000, 2003, 2012; Balusu et al, 2016a; Gorle et al, 2016). Moreover, Chalbot et al (2011) pointed out that CPE damage may be one of the first signs of AD. In agreement with these observations, we observed that Aβ oligomers alter the cuboidal shape of CPE cells. Also in the APP/PS1 model, we observed a high degree of morphological and cellular

damage in the choroid plexus. Notably, these pathological changes in the choroid plexus were prevented by the absence of TNFR1 in both models. We previously reported that icv injection of AβO increases the leakiness of the blood–CSF barrier associated with decreased expression of tight junction genes and proteins, and increased activity of MMPs (Brkic et al, 2015b). Tight junctions are essential for maintaining the integrity of several epithelial and endothelial barriers, including the blood–CSF barrier. Their apical presence in CPE cells restricts paracellular diffusion of molecules through this barrier (Redzic, 2011; Zihni et al, 2016). MMPs, proteases that degrade components of the extracellular matrix, are involved in several neurodegenerative disorders (Brkic et al, 2015a). Additionally, we and others have shown that MMPs can also disrupt structural tight junction proteins of the BBB and blood–CSF barrier (Yang et al, 2007; Schubert-Unkmeir et al, 2010; Vandenbroucke et al, 2012). Importantly, the choroid plexus is considered as a selective entry gate for leukocytes into the CNS and essential for immunosurveillance of the CNS. Consequently, changes in the choroid plexus anatomy and integrity of the barrier might allow more leukocyte extravasation into the brain parenchyma, which would exacerbate CNS inflammation (Shrestha et al, 2014; Demeestere et al, 2015).

In our previous study, we speculated that these detrimental events in the choroid plexus are caused by a cascade of upstream pro-inflammatory mediators (Brkic et al, 2015b). Here, we confirmed this hypothesis by showing that TNFR1 signaling is the main pathway involved in these detrimental events in the choroid plexus. Indeed, we confirmed that TNFR1 deficiency prevents the increase in blood–CSF barrier permeability because MMP expression in the choroid plexus is not increased and the choroid plexus tight junctions are preserved. Bergen et al (2015) pointed out that Cldn5 is the main gatekeeper for paracellular transport across the blood–CSF barrier and that its down-regulation is responsible for the blood–CSF barrier leakage. As TNFR1 deficiency preserves *Cldn5* expression in the choroid plexus after AβO triggering, this might be an important reason why the blood–CSF barrier is preserved in TNFR1-deficient mice.

MMP3 is designated as the most important MMP responsible for barrier leakage in AβO-induced toxicity (Brkic et al, 2015b), and its expression as well as that of *Mmp8* is reduced in TNFR1 null mice. This might have a substantial effect because MMP8 is known to impair the blood–CSF barrier in inflammatory conditions (Vandenbroucke et al, 2012). Here, we identified TNF signaling via TNFR1 as the main regulator of AβO-induced damage to the choroid plexus and the barrier. Whether or not TNFR1 abrogation also prevents leukocyte recruitment to the CNS is not yet known and should be addressed in future studies.

We also observed that TNFR1 deficiency might do more than protect the CNS against inflammatory triggers from outside the CNS by preserving the blood–CSF barrier; it might also impair the activation of resident microglia. Importantly, AβO induces the release of TNF from microglia, which then stimulates it own release via TNFR1 (Kuno et al, 2005). Thereby, a vicious feedback loop in the activation of microglia is sustained, but this loop is broken by TNFR1 deficiency.

A dual role for primed microglia has been described. In early AD onset, microglia may contribute to clearance of Aβ by phagocytosis, but later they initiate deleterious cascades by releasing pro-inflammatory cytokines, which contribute to neuronal damage (Agostinho et al, 2010; Yang et al, 2011; Wang et al, 2015).

Interestingly, it has been shown that although peripheral human TNF could modulate Aβ pathology by increasing the phagocytic activity of microglia, it contributes to neuronal damage and synaptic loss as well (Paouri *et al*, 2017a). In contrast to these results, it has also been shown that TNF inhibits phagocytosis of Aβ by microglia (Koenigsknecht & Landreth, 2004). As we observed that TNFR1 ablation prevented the activation of microglia, this might abrogate the sustained neuroinflammation and prevent neuronal and synaptic loss. Because microglia are involved in Aβ clearance, the observed reduction in activated microglia might seem to contradict our observations in APP/PS1$^{tg/wt}$TNFR1$^{-/-}$ mice, which displayed a lower Aβ burden and less plaque deposition. However, it has been shown that in some conditions activated

microglia fail to clear Aβ despite their abundance (Koenigsknecht & Landreth, 2004). In our study, the reduced Aβ burden could have been a consequence of the down-regulation of the BACE1 promoter activity rather than to increased clearance by microglia and this is in agreement with the study of He *et al* (2007). Additionally, Paouri *et al* recently found that TNF abrogation attenuates amyloid plaque formation by restricting the activity of also PS1, one of the four core proteins in the γ-secretase complex responsible for APP processing, in 5xFAD mice (Paouri *et al*, 2017b), but the involvement of TNFR1 in this process is not investigated.

Finally, the most promising finding of our study is that TNFR1 deficiency alleviated cognitive decline in both AD models. Moreover, TNFR1 inhibition by a Nanobody recapitulated these findings

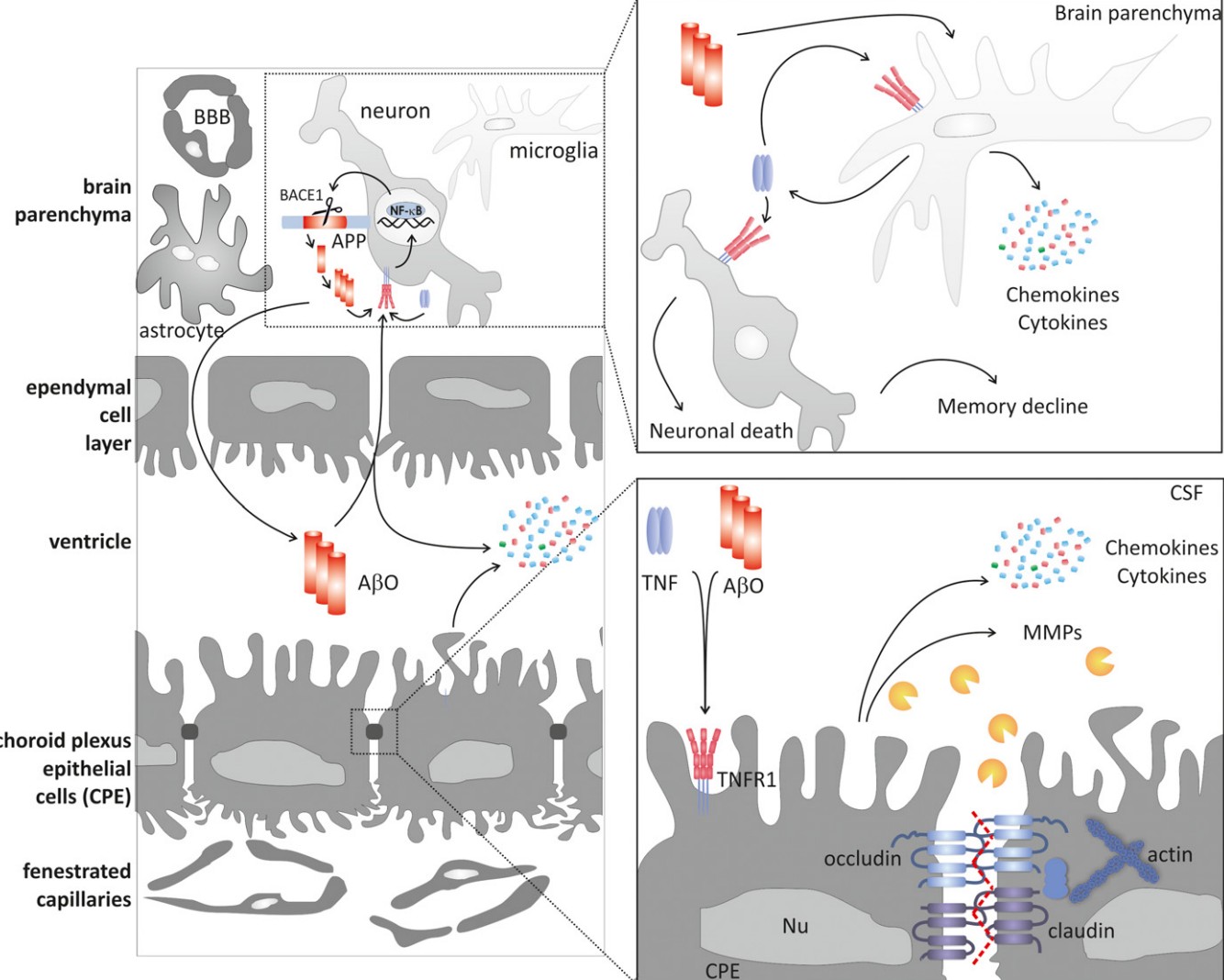

**Figure 10.  Schematic hypothesis of the roles of TNFR1 in the pathology of Alzheimer's disease.**
Amyloid precursor protein (APP) is cleaved into pathogenic amyloid-beta 1–42 (Aβ$_{1-42}$) species by BACE1, the expression of which can be activated by TNFR1-induced NF-κB in neurons of the hippocampus. Choroid plexus epithelial (CPE) cells are tightly connected with tight junctions. The presence of Aβ oligomers (AβO) in the CSF leads, via TNFR1, to induction of chemo- and cytokines (e.g., TNF) by CPE cells and other neuronal cells such as microglia and neurons. Upon AβO exposure, CPE cells also express matrix metalloproteinases (MMPs), which disrupt tight junctions (right lower panel). Eventually, this results in leakage of the blood–CSF barrier. In the brain parenchyma, TNF triggers TNFR1 molecules on the surface of microglia and consequently induces its own release. Additionally, TNF also promotes neuronal death via TNFR1. Ultimately, all these events lead to further aggravation of neuroinflammation and to cognitive impairment.

                                                                            

by protecting against AβO-induced memory impairment. Although this could have been a consequence of the reduction in neuroinflammation and amyloid burden, it can also be a direct effect of modulation of the synapses. Indeed, TNF has been shown to induce the release of glutamate from microglia and astrocytes via TNFR1 signaling and might directly induce glutamate excitotoxicity via TNFR1 through activation of AMPA receptors (Bezzi et al, 2001; Kuno et al, 2005; Takeuchi et al, 2006). Conversely, TNFR2 activation protects against glutamate-induced excitotoxicity (Marchetti et al, 2004; Olmos & Llado, 2014). Moreover, it has been shown that selective inhibition of soluble TNF restores the synaptic dysfunction and long-term potentiation impairment in 5XFAD mice (MacPherson et al, 2017). Further research is needed to address these possibilities in our established models.

In conclusion, we propose new mechanisms through which TNFR1 might contribute to AD neuroinflammation and pathology, in addition to its direct effect on neuronal cell death (Yang et al, 2002b; Li et al, 2004; Fig 10). TNF/TNFR1 signaling leads to neuroinflammation in the choroid plexus and the hippocampus, contributing to disruption of choroid plexus morphology and blood–CSF barrier integrity. This may result in aggravation of CNS inflammation and eventually to neuronal loss. Additionally, TNFR1 deficiency prevents microglia activation, which results in reduced inflammation of the brain parenchyma. We have also shown that impairment of TNF/TNFR1 protects AD-susceptible mice against Aβ-induced pathology. Finally, both genetic and pharmacologic abrogation of TNFR1 signaling rescued AD-associated cognitive impairment, which is the ultimate consequence of all the destructive events in AD pathology.

# Materials and Methods

## Mice

For the experiments with intracerebroventricular (icv) injections of oligomerized Aβ$_{1-42}$ (AβO), female C57BL/6J wild-type (WT) and C57BL/6J TNFR1 knockout (KO) (TNFR1$^{-/-}$) mice, aged 8–12 weeks, were used and were housed in SPF conditions. B6-Tg(Thy1-APPswe; Thy1-PS1 L166P) in a C57BL/6J background, further called APP/PS1$^{tg/wt}$, corresponding non-transgenic littermates (APP/PS1$^{wt/wt}$) and APP/PS1$^{tg/wt}$ mice in a TNFR1$^{-/-}$ background were bred in a conventional facility. TNFR1$^{-/-}$ mice are described by Rothe et al (1993), and the APP/PS1 mouse model is developed by Radde et al The latter mice harbor the human APP transgene bearing Swedish mutation (KM670/671NL) and PSEN1 L166P under control of Thy1 promoter (Radde et al, 2006). APP/PS1 mice were crossed into a TNFR1 KO background. In all experiments, we used heterozygous transgenic littermates that were age- and gender-matched. All mice were housed with 14- to 10-h light and dark cycles and free access to food and water. All experiments comply with the current laws of Belgium and were approved by the animal ethics committee of Ghent University.

## Ingenuity pathway analysis

Human choroid plexus samples of AD patients and healthy controls were obtained post-mortem at autopsy from patients with advanced late-stage Alzheimer's disease [Braak & Braak stage V–VI, mean age

79 years, 95% confidence interval (CI) (72.2–86.4)] and from healthy controls [mean age 58 years, 95% CI (41.9–72.9)]. Samples were immediately snap-frozen upon removal in liquid nitrogen. All samples were obtained from 3.5 to 30 h post-mortem. All tissues were stored at −80°C at the Brown University Brain Tissue for Neurodegenerative Disorders Resource Center until processed. This research application was reviewed and approved as meeting the standards for the protection of human subjects per 45CFR46/21CFR56 by Rhode Island Hospital's Committee on protection of human subjects. The IRB for the Brain Bank falls under Project #0083-03. Since this is a post-mortem de-identified study, the IRB decided that no formal review was required (IRB Registration numbers: RIH IRB 1-0000396, RIH IRB 2-00004624, TMH IRB-00000482). Gene expression was analyzed by Ingenuity pathway analysis (IPA) (Ingenuity® Systems, www.ingenuity.com). A core analysis was performed, and networks, canonical pathways, and functional enrichment in diseases and gene function were determined. In our analysis, we mainly focused on cytokines. Confirmed relationships (e.g., TarBase, IPA expert findings) and relationships predicted (TargetScan) with at least moderate confidence, as stated in IPA, were both accepted.

## Preparation of Aβ$_{1-42}$ oligomers

Aβ$_{1-42}$ oligomers (AβO) were prepared as previously described (Brkic et al, 2015b). Briefly, Aβ$_{1-42}$ (rPeptide; A-1163-1) or scrambled peptide (rPeptide; A-1004-1) was dissolved in hexafluoroisopropanol (HFIP; Sigma-Aldrich; 105228) at 1 mg/ml. Next, HFIP was removed with a SpeedVac vacuum concentrator. The resulting peptides were then dissolved in DMSO and purified using a 5 ml HiTrap desalting column (GE Healthcare; 17-408-01). The monomeric Aβ$_{1-42}$ was eluted with Tris-EDTA buffer (50 mM Tris and 1 mM EDTA, pH 7.5). The eluted peptide concentration was measured using Pierce Micro BCA Protein Assay (Thermo Scientific; 23225) according to the manufacturer's guidelines. The peptides were allowed to oligomerize at room temperature (RT) for 2 h, followed by dilution to 1 μg/ml or 2 μg/ml in Tris-EDTA buffer. All icv injections were performed within 2 h after oligomerization.

## Intracerebroventricular (icv) injections

For the icv injections of scrambled peptide, AβO or AβO with Armenian hamster anti-TNFR1 antibody (Ab) or TROS, mice were anaesthetized with isoflurane and mounted on a stereotactic frame. A constant body temperature was maintained using a heating pad. Injection coordinates were determined from the bregma (anteroposterior 0.07, mediolateral 0.1, dorsoventral 0.2) and were determined using the Franklin and Paxinos mouse brain atlas. By using a Hamilton needle, 5 μl of the either scrambled peptide or AβO (1 μg/ml) was injected into the left lateral ventricle. For co-injections, 5 μl was injected of a mixture of 2.5 μl AβO (2 μg/ml) and 2.5 μl of the trivalent anti-TNFR1 Nanobody TROS [1.55 mg/ml, (Steeland et al, 2015)].

## Cerebrospinal fluid isolation

Cerebrospinal fluid (CSF) was harvested from the fourth ventricle via the cisterna magna puncture method (Brkic et al, 2015b). Briefly, borosilicate glass capillary tubes (Sutter Instruments; B100-75-15) were used to make needles on the Sutter P-87 flaming

micropipette puller (pressure 330 Pa, heat index 300). Mice were anaesthetized, and the skin from the backside of the mice was dissected and disinfected with 70% ethanol. The cisterna magna membrane was exposed by separating the muscle tissue on the back side of the scull, and the animal was mounted at an angle of 135°. Subsequently, CSF was collected by puncturing the cisterna magna membrane using the capillary needles.

### Determination of blood–CSF barrier permeability leakage

One hour before CSF isolation, mice were intravenously (iv) injected with 250 mg/kg FITC-labeled dextran (4 kDa, Sigma). A CSF sample of 2 μl was diluted 50 times in PBS, and leakage from the blood into the CSF was determined by measuring the fluorescence in the diluted CSF samples ($\lambda_{ex}/\lambda_{em}$ = 485/520 nm) using the FLUOstar omega (Isogen LifeScience).

### Tissue sample isolation

For RNA analysis, mice were transcardially perfused with a mixture of 500 ml DPBS (Sigma), 1 ml heparin sodium (5,000 IU/ml, Wockhardt), and 0.5% bromophenol blue (Sigma) to remove all blood and to label the choroid plexus for easy isolation. Choroid plexus and hippocampus were isolated from brain tissue and snap-frozen in liquid nitrogen for RNA analysis. For immunohistochemical analysis, mice were transcardially perfused with paraformaldehyde (PFA) and brain samples were fixed in 4% PFA and processed for paraffin-embedding.

### Immunohistochemistry

For immunostaining on mouse brain sections, 5-μm sections were prepared from paraffin-embedded samples. After dewaxing, samples were treated with citrate buffer (Dako; S2031), followed by washing in PBS. Endogenous peroxidase activity was blocked with 3% $H_2O_2$ in methanol for 10 min, and samples were blocked with 5% goat serum in antibody diluent (Dako; S2022) for 30 min at RT, followed by overnight incubation at 4°C with primary antibody against claudin-1 (CLDN1, Invitrogen, 51-9000, 1:1,000) or IBA1 (Wako Chemicals, 019-19741, 1:500). The next day, slides were washed with PBS and incubated with secondary antibody coupled to HRP or biotin, respectively (Dako; EK4003 and E0432). An amplification step was performed using tyramide (TSA kit, Perkin Elmer) for CLDN1-staining, and visualization was done using ABC (Vector) and DAB. Slides were counterstained with hematoxylin, dehydrated, and mounted with Entalan. Slides were scanned using a slide scanner (ZEISS, Axio Scan) and analyzed with the Zen software (Carl Zeiss Microscopy GmbH, 2012). Histological quantification of the plaques in the whole brain was done with the Fiji software (University of Wisconsin-Madison) by color thresholding the brown color, using the exact same parameters for all genotypes.

### Transmission electron microscopy

The choroid plexus of the fourth ventricle was dissected from APP/PS1 mice of 18-week-old in WT or TNFR1$^{-/-}$ background, or from WT or TNFR1$^{-/-}$ mice 6 h after injection of AβO, and these were fixed in 4% PFA and 2.5% glutaraldehyde in 0.1 M NaCacodylate

buffer, pH 7.2 for 4 h at RT followed by fixation O/N at 4°C. After washing in buffer, they were post-fixed in 1% $OsO_4$ with 1.5% $K_3Fe$ $(CN)_6$ in 0.1 M NaCacodylate buffer at RT for 1 h. After washing in dd$H_2O$, samples were subsequently dehydrated through a graded ethanol series, including a bulk staining with 1% uranyl acetate at the 50% ethanol step followed by embedding in Spurr's resin.

Ultrathin sections of a gold interference color were cut using an ultramicrotome (Leica EM UC6), followed by a post-staining in a Leica EM AC20 for 40 min in uranyl acetate at 20°C and for 10 min in lead stain at 20°C. Sections were collected on formvar-coated copper slot grids. Grids were viewed with a JEM 1400plus transmission electron microscope (JEOL, Tokyo, Japan) operating at 60 kV.

### 3D scanning electron microscopy

For 3D scanning electron microscopy (SEM), the choroid plexus of the lateral ventricle was dissected from APP/PS1 mice of 18-week-old in WT or TNFR1$^{-/-}$ background, or from WT or TNFR1$^{-/-}$ mice 6 h after injection of AβO and scrambled peptide, and was immediately transferred into fixation buffer (2% PFA, Sigma-Aldrich; 2.5% glutaraldehyde, Electron Microscopy Sciences in 0.15 M Cacodylate buffer, pH 7.4). After overnight fixation at 4°C, samples were washed 3 × 5 min in Cacodylate buffer and subsequently osmicated in 2% osmium (EMS), 1.5% ferrocyanide, and 2 mM $CaCl_2$ in Cacodylate buffer for 1 h on ice, and then extensively washed in ultrapure water (UPW). This was followed by incubation in 1% thiocarbohydrazide (20 min), washes in UPW, and a second osmication in 2% osmium in UPW (30 min). The samples were washed 5 × 3 min in UPW and placed in uranyl acetate replacement stain (Electron Microscopy Sciences), 1:3 in water at 4°C overnight. The following day, they were stained with Walton's lead aspartate stain for 30 min at 60°C. For this, a 30 mM L-aspartic acid solution was used to freshly dissolve lead nitrate (20 mM, pH 5.5). The solution was filtered after incubation for 30 min at 60°C. After the final washes, the samples were dehydrated using a series of ice-cold solutions of increasing EtOH concentration (30, 50, 70, 90, 2 × 100%), followed by two dehydrations of 10 min in propylene oxide. Subsequent infiltration with resin (Spurr's; EMS) was done by first incubating the samples overnight in 50% resin in propylene oxide, followed by at least two changes of fresh 100% resin. Next, samples were embedded in fresh resin and cured in the oven at 65°C for 72 h.

For serial block-face imaging, the resin-embedded samples were mounted on an aluminum specimen pin (Gatan) using conductive epoxy (Circuit Works). The specimens were trimmed in a pyramid shape using an ultramicrotome (Ultracut; Leica), and the block surface was trimmed until smooth and at least a small part of tissue was present at the block face. Next, samples were coated with 5 nm Pt in a Quorum Q 150T ES sputter coater (Quorum Technologies). The aluminum pins were placed in the Gatan 3View2 in a Zeiss Merlin SEM for imaging at 1.6 kV with a Gatan Digiscan II ESB detector. The Gatan 3View2 was set to 300 sections of 70 nm. IMOD (http://bio3d.colorado.edu/imod/) and Fiji (Schindelin *et al*, 2012) were used for registration of the 3D image stack and conversion to TIFF file format. Representation of the cell in 3D movies and snapshots was done in Imaris (BitPlane).

## Detection Aβ plaques

Plaques of Aβ in the brain were detected with a Thioflavin-S staining (binds β-rich structures). Thioflavin-S binds senile plaques—extracellular Aβ deposits—and neurofibrillary tangles, the two characteristic lesions in AD patients. Aβ is only present in senile plaques, and thus, the antibody only detects the senile plaques. For the Thioflavin-S staining (Sigma; T1892), PFA-fixed brain sections of 8–10 μm were prepared from paraffin-embedded samples. After dewaxing, sections were incubated with filtered 0.05% aqueous Thioflavin-S (0.1 g Thioflavin-S in 200 ml 50% ethanol, Sigma; T1892) for 5 min at RT. Next the sections were washed with 70 and 95% ethanol for 5 min, followed by three washing steps in PBS for 5 min. Next, the slides were incubated with DAPI (1:2,000) in PBS for 30 min at RT. After washing with PBS and distilled $H_2O$, the slides were mounted with Mowiol-containing mounting medium. Slides were scanned using a slide scanner (ZEISS, Axio Scan) and analyzed with the Zen software (Carl Zeiss Microscopy GmbH, 2012). Histological quantification of the plaques in the whole brain was done with the ImageJ software (https://imagej.net) by thresholding the Thioflavin-S staining, using the exact same parameters for all genotypes. Plaque burden was calculated as the area occupied by the plaques, determined by the ImageJ particle counting plug-in, divided by the total brain region area.

## Aβ extraction and ELISA

Both the soluble and insoluble Aβ fractions—extracted from the cortex—were determined with ELISA. For the extraction, tissue was homogenized in Tissue Protein Extraction Buffer (10 μl per μg cortex; Thermo Scientific) supplemented with protease inhibitor (Thermo Scientific) and phosphatase inhibitor tablets (Sigma-Aldrich) using a Precellys (Bertin Technologies). The beads in the homogenized samples were spin down for 5 min at 5,000 *g* and 4°C. Supernatant was collected and centrifuged at 4°C for 1 h at 100,000 *g* (TLA 100 rotor). Supernatant containing the soluble Aβ fraction could directly be used for ELISA, while the pellet containing insoluble Aβ is further processed. 2 μl GuHCl solution (6 M GuHCl + 50 mM Tris–HCl) supplemented with protease inhibitor, per μg cortex from which initially was started, was added to the pellet containing the insoluble Aβ fraction. The mixture was sonicated at following program 10 cycli 30 s on—30 s off, high amplitude (Bioruptor™ Sonication Device), and then, the mixture was vortexed for 5 min and incubated for 60 min at 25°C. The mixture was centrifuged at 4°C for 20 min at 70,000 *g*. The supernatant was 12 times diluted with GuHCl diluent (0.26 g $NaH_2PO_4.2H_2O$, 0.41 g $Na_2HPO_4.2H_2O$, 16 ml 5 M NaCl, 0.8 ml 0.5 M EDTA, 1 ml 10% $NaN_3$, 1.5 ml 10% CHAPS in 200 ml distilled $H_2O$, pH 7, supplemented with 10% BlockAce, 0.4 g BSA, and 1 protease inhibitor tablet (Roche) prior to use) to prepare the samples for ELISA.

To determine the $Aβ_{1-42}$ and $Aβ_{1-40}$ levels extracted out of the cortex, 96-well immunoplates (Maxisorp Nunc; 430314) were coated overnight (ON) at 4°C on a shaker with 50 μl per well anti-$Aβ_{42}$ antibody (1.5 μg/ml; JRF/cAb42/26) or anti-$Aβ_{1-40}$ antibody (1.5 μg/ml; JRF/cAb40/28) in coating buffer (10 mM Tris–HCl, 10 mM NaCl, 10 mM $NaN_3$ in 500 ml distilled $H_2O$, pH 8.5). Plates were washed five times with PBST (PBS + 0.05% Tween-20), and residual protein binding sites were blocked for 4 h at RT with 100 μl blocking buffer (0.1% casein buffer). 30 μl of either standard ($Aβ_{1-42}$; rPeptide or $Aβ_{1-40}$; rPeptide) or sample was mixed with 30 μl detection antibody (JRF/ABN/25 coupled to HRPO (Janssen Pharmaceutica), 1:2,000, diluted in blocking buffer). After blocking, ELISA plates were washed five times with PBST and 50 μl of the standard/sample-detection mixtures was added to the ELISA plates. Plates were incubated ON at 4°C, while slowly shaking. Absorption at 450 nm was measured after adding 50 μl substrate solution [100 μl TMB (BD Biosciences OptEIA™)] followed by stopping buffer (50 μl 1 M $H_2SO_4$). The amount of $Aβ_{1-42}$ was determined with GraphPad Prism 7.0 using a nonlinear regression model.

## Novel object recognition test

Cognitive behavior was assessed by the basic protocol for the NOR test described by Antunes and Biala (2012). The NOR test is based on the innate tendency of mice to differentially explore novel objects over familiar ones. Therefore, mice were placed into the testing room which is a rectangular, clear open-field area made from acrylic (40 × 40 × 40 cm). The procedure includes three phases: habituation, training, and testing. Habituation was performed in the prior day when the animals were positioned into the open-field arena without any object to familiarize with the environment for 5 min. The training phase was performed the day after the habituation. Mice were placed in the arena facing the wall opposite to new identical objects which were put at two opposite positions in the box at the same distance from the nearest corner. Mice were allowed to freely explore the two objects (A1 and A2) for 5 min and were then returned to their home cage. In the testing phase, memory was tested either 15 min (STM) or 24 h (LTM) after the training session. Mice were placed back in the same box after replacing one of the familiar objects (A1 or A2) by a novel one (A3 or A4), and the 5-min testing phase was started. The exploration of the objects was considered as deliberate contact with their mouth or nose that occurred with each object. The exploration time for the familiar or the new object during the test phase was recorded. The percentage preference for the novel object was calculated using the following formula: % novel object preference = [novel object exploration time/(novel object exploration time + familiar object exploration time)] × 100%. A value of 50% reflects no preference for any of the objects, and a value > 50% indicates preference for the novel object. To exclude the existence of olfactory clues, all objects were thoroughly cleaned with 20% ethanol after each trial.

## Real-time qPCR

RNA was isolated from choroid plexus and hippocampus using the Aurum total RNA Mini Kit (Bio-Rad). RNA concentration was measured using the Nanodrop 1000 (Thermo Scientific), and cDNA was synthesized by the iScript cDNA Synthesis Kit (Bio-Rad). qPCR was performed with the Light Cycler 480 system (Roche) using SensiFast SYBR No-Rox (Bio-Line). Expression levels were normalized to the expression of the two or three most stable housekeeping genes, which were determined for each organ and condition using geNorm (Vandesompele *et al*, 2002): APP/PS1

mice choroid plexus: *Gapdh* and *Rpl*; hippocampus: *β-actin*, *Hprt*, *Rpl*. AβO injections in WT or TNFR1$^{-/-}$ mice, choroid plexus, and hippocampus: *Gapdh*, *Hprt*, *Rpl*. The primer sequences of the forward and reserve primers for the different genes are provided in Appendix Table S1.

## Statistics

Statistics were performed using GraphPad Prism 7.0 (GraphPad Software, Inc.). Bars represent mean ± SEM. qPCR data were analyzed with an unpaired *t*-test unless mentioned differently. Permeability data and data concerning short-term and LTM data were compared with one-way ANOVA. Significance levels are indicated $*0.01 \leq P < 0.05$; $**0.001 \leq P < 0.01$; $***0.001 \leq P < 0.0001$; and $****P < 0.0001$. Exact *n*- and *P*-values of the significant results can be found in Appendix Table S2.

## Data availability

The choroid plexus microassay data from this publication have been deposited to the GEO database (https://www.ncbi.nlm.nih.gov/geo/) and assigned the identifier GSE110226.

## Ethical approval

All procedures performed in the studies involving animals were performed in accordance with the ethical standards of the institution at which the studies were conducted and of that of the Belgian law.

**Expanded View** for this article is available online.

## Acknowledgements

We thank Joke Vanden Berghe for technical assistance related to mice breeding and Sara Van Ryckeghem and Bavo Vanneste for technical assistance with sample processing. We want to thank Amin Bredan for editing the manuscript and Peter Borghgraef for the sample preparation for SBF-SEM. M. Mercken (Johnson & Johnson Pharmaceuticals Research and Development, Beerse, Belgium) and Bart De Strooper (VIB-KULeuven) are acknowledged for providing the antibodies against Aβ (JRF/ABN/24, JRF/cAB40/28, and JRF/cAB42/26) for the AβO ELISA and assistance to perform the AβO ELISA procedure. This work was supported by the Research Foundation–Flanders (FWO), the Concerted Research Actions of Ghent University, The Foundation for Alzheimer's Research Belgium (SAO-FRA) and EU Cost action MouseAge (BM1402), and the Baillet Latour Fund. S.S., N.G., and C.V are PhD students that are funded by the Research Foundation–Flanders (FWO). The Zeiss Merlin with Gatan 3View2XP was acquired through a CLEM grant from Minister Ingrid Lieten to the VIB Bio-Imaging Core.

## Author contributions

SS and REV designed the study. SS, NG, CV, and MB performed the mouse experiments. GVI, CVC, EVW, SB, and REV assisted with experiments and sample processing. RDR prepared the samples for TEM and helped with the imaging and interpretation. AK and SL prepared the samples for serial block-face scanning electron microscopy (SBF-SEM) and performed image acquisition. ES and CEJ provided the microarray data from human choroid plexus. SS and REV analyzed and interpreted the data. SS, CL, and REV drafted the manuscript. All authors revised the manuscript critically for important intellectual content and approved the final version.

---

### The paper explained

**Problem**

Alzheimer's disease (AD) is a devastating neurodegenerative disorder that is the leading cause of dementia in patients older than 65 years. Due to the aging of the population, the number of AD patients is expecting to rise to 130 million by 2050. Currently, AD is the only cause of death among the top 10 in the USA without medication to prevent, delay, or cure the disease, except for some drug that mask the symptoms or temporarily delay worsening of the symptoms. Therefore, identifying new therapeutic targets by elucidating molecular mechanisms of the pathology are indispensible for the development of new drug.

**Results**

We report that the TNF/TNFR1 signaling pathway is upregulated in the choroid plexus of late-stage AD patients. We confirmed the detrimental role of this pathway in two murine models of AD and we provide evidence that the interaction of TNF with TNFR1 is responsible for the morphological alterations observed in the CPE cells. We found that genetic abrogation of TNFR1 protects against amyloid-beta (Aβ)-induced impairment of the blood–cerebrospinal fluid barrier by reducing the inflammatory environment. TNFR1 deficiency also leads to a reduction in microglia count and prevents amyloidogenesis. Genetic and pharmacological blockage of TNFR1 impedes memory decline typical for this disease.

**Impact**

This study addresses the largest unmet medical need in neurology. TNF/TNFR1 signaling in the choroid plexus is as novel pathway contributing to AD neuroinflammation, and we identified this pathway as a novel therapeutic drug target. Importantly, a TNFR1-targeting Nanobody supports this idea as this drug substantially improved the cognitive decline that is associated with the disease. Future research is needed to address the possibilities of this Nanobody in established AD and in patients.

---

## Conflict of interest

The authors declare that they have no conflict of interest.

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
