## [Review Process File · EMBO Molecular Medicine]

Counteracting the effects of TNF receptor-1 has therapeutic potential in Alzheimer's disease

Sophie Steeland, Nina Gorié, Charysse Vandendriessche, Sriram Balusu, Marjana Brkic, Caroline Van Cauwenberghe, Griet Van Imschoot, Elien Van Wonterghem, Riet De Rycke, Anneke Kremer, Saskia Lippens, Edward Stopa, Conrad E. Johanson, Claude Libert, Roosmarijn E. Vandenbroucke

Review timeline:

Submission date:	21 July 2017
Editorial Decision:	15 August 2017
Revision received:	15 November 2017
Editorial Decision:	12 December 2017
Revision received:	31 December 2017
Editorial Decision:	08 January 2018
Revision received:	10 January 2018
Accepted:	17 January 2018

Editor: Céline Carret

Transaction Report:

1st Editorial Decision

15 August 2017

Thank you for the submission of your manuscript to EMBO Molecular Medicine. We have now heard back from the two referees whom we asked to evaluate your manuscript.

You will see from the set of comments below that both referees find the data interesting, however both also highlight inadequacies. Of particular relevance, we feel that the study would be strengthened by comparing/correlating the acute to the chronic models to better reflect the clinical relevance, and providing an appropriate nanobody control to ascertain the causative effect. In addition, data presentation and discussions as well as missing details should be improved and provided as suggested.

We would therefore welcome the submission of a revised version within three months for further consideration. Please note that EMBO Molecular Medicine strongly supports a single round of revision and that, as acceptance or rejection of the manuscript will depend on another round of review, your responses should be as complete as possible.

I look forward to receiving your revised manuscript.

***** Reviewer's comments *****

Referee #1 (Comments on Novelty/Model System):

The authors have been diligent in using both the APP/PS1 model and ICV injection of AbO but there are what might be considered rather extreme responses to the acute model and the disease relevance of the doses of AbO require more contextualisation. In particular in the section of choroid plexus epithelial damage, it is really not clear whether such an injury arising from an acute ICV injection is of real relevance to AD pathology.

Referee #1 (Remarks):

In this study the Vandenberg group study the role of TNF signaling in various aspects of AD-related pathology. The work is generally of high quality and the authors go to some lengths to cover different approaches to studying events likely to occur in the disease: human material, APP/PS1 mice and Amyloid-beta oligomers. This is more than most groups do and the authors should be encouraged for this approach. Nonetheless, some of the key questions arise precisely because of the difficulty in comparing what happens in the 'chronic' versus the 'acute' model.

Methods

Human choroid plexus were said to be snap frozen...were they obtained directly at post-mortem. What was the delay from death to collection?

Figure 1

To clarify, in Figure 1a: TNF appearing as the top hit does not imply that TNF was the most highly expressed, but that a large panel of genes that share TNF-signalling as a common inducer make TNF-signalling the top hit? Likewise for IL-1 as the second hit? Figure 1b is presented at a size too small for genes to be legible.

Figure 2.

It is appropriate that the authors note that the addition of A β O is an acute neurotoxicity model. However, the acute nature of this stimulus is very apparent in the very large responses in the choroid plexus and hippocampus compared to the APP/PS1 mice. That choroid plexus responses are 10-100 fold higher than in the APP/PS1 mice should be made explicit in the text. Moreover, the concentration of AbO injected ICV should be explicit in the results text or the legend and some attempt to compare these doses to levels of Ab oligomers in the APP/PS1 brain should be made. The magnitude of the inflammatory transcript increases seem excessively high and it highlights the need for genuine discussion of the appropriateness of acute AbO injection as a model of anything that is happening in Alzheimer's disease. Presumably these oligomers are being made all the time in the brain tissue of the AD patient and the APP/PS1 mice? Why do the hippocampus and the choroid plexus produce 2 fold and no increase respectively if these oligomers robustly induce TNF-a?

Figure 3

Much of the labeling of axes is infuriatingly small even when replicated at full printed page size. The animals/treatments on the X axis are illegible. Some of these comments may suffer from misconceptions based on my inability to read the axes, but figure 3f (IL-1 in HPC) does not seem consistent with the decreased IL-1b expression presented in Figure 2b. For the AbO, blocking TNF signaling abrogates most or all inflammatory responses.

Figure 4, 5

The effects of AbO choroidal epithelium are very impressive, and are prevented in TNFR1^{-/-} mice. However, the AbO dose once again becomes important to contextualize with respect to AD-associated levels of similar oligomers. The authors report morphological change of the CP in AD patients and also report changes to epithelial cell morphology in their prior AbO experiments. However, they make no attempt to compare/contrast these alterations to the CP in their acute model to any changes in AD patients and do not mention what happens to the CP in APP/PS1 mice. It would be extremely useful to show to what extent there are similar changes in the CP of APP/PS1 mice. Of course they are likely to be much less obvious and this is part of the problem: it once again raises the hard-to-ignore feeling that AbO provides a highly exaggerated version of amyloid-induced pathological changes

Figure 6

Increased dextran leakage into the CSF occurs after AbO and this should be more explicit in the way that Fig 6a is labeled (on the axes) or explained (in the legend). The reader should not have to return to the methods section to interpret what this graph means. In (f) the labels are, once again, hard to read. It looks, to this reader as though the cln1 tight junctions are more intact in the AbO TNFR1^{-/-} mice than in the scrambled Ab group. I do see that the pictures are annotated with arrows and arrowheads but the patterns look much more similar between the left and centre than between the left and right. How robust is this measure? Even the areas chosen to compare are quite different (ie far less CP in the left image).

Figure 7

Several graphs of amyloid quantification are presented but none of the raw data are visible. For example pictures of thioflavin S labeling should be shown in relevant experimental groups. The text reflects 'significantly fewer' plaques of various sizes but this refers to statistical significance: they should also reflect the fairly modest decreases (ie 15-20% reduction). The statistics that have been carried out are only partially covered in the figure legend, with no information for c and d and a descriptor suggesting that there are panels e-j (obviously a relic of a prior version). It is not clear why cognitive impairments in the APP/PS1 model are tagged on here but cognitive deficits in the AbO model are tagged onto IBA-1 labelling that is shown for both models. If I understand the bar and min-max box plots, the values annotated and included in the text do not look right: In fig 7i the left bar is said to be 64.18 but the bar looks to be below 60, while in the middle bar, it is said to be 50.8 but the bar looks more like 55. This requires some explanation (and is of pressing relevance to key result).

Figure 8

The IBA-1 pictures are inadequate to show what the graphs refer to. Is brown staining really quantified in the 'whole brain section'? Why are pictures shown for the AbO-injected but not the APP/PS1? What is the quality of the IBA-1 labelling? With such a clear effect of TNFR1 deficiency on microglial-positive area one would really like to see this in some pictures. Why would the authors choose to image 'whole brain section': why not representative areas in the hippocampus and cortex so that cells can be examined to analyse the pattern of microgliosis within these areas? If the images in c represent an 'area close to the choroid plexus', why is the choroid plexus not shown and why does the middle image not contain the (presumed) ventricle that is visible in the other 2 pictures? The data presented in the graphs look very robust and the images to support this really should be shown. For quantitation of images to be credible the reader should be able to examine sample images to increase confidence in the numbers generated from these images. As highlighted for fig 7i above, in figure 8e the middle bar is said to be 49.99 but the bar looks to be more like 53 to 54. Again, this requires some explanation and is relevant for the key result.

There are some similar issues to address in figure 9.

The latter part of the paper feels a little disorganised, with the cognitive data distributed in a way that does not make obvious sense to this reader. I think it could be significantly improved by taking time to present the amyloid more clearly and with pictures, in a figure on its own. Then doing likewise for IBA-1, again showing pictures in both models. Finally the NOR task could be presented for all three models in a third figure - but with more attention to the detail of the box plots.

I would also wonder why, after so much focus on the choroid plexus early in the paper, there is not more attention on implicating the choroid plexus changes in the cognitive impairments ultimately observed. It gives a sense of discontinuity despite the impressive effects of TNF disruption on almost every measure examined. Which ones are most important for cognitive dysfunction is a key question and whether that parameter would stand up across the multiple models would obviously be important to establish.

Referee #2 (Comments on Novelty/Model System):

The key experiment lacks an adequate control, see statement below. Other than that, the technical quality of the work is good.

Referee #2 (Remarks):

In the present manuscript Steeland et al. report on signaling activity of TNFR1 that might contribute to manifestation of Alzheimer's Disease (AD). Interfering with this signaling activity appears to attenuate the pathophysiological and neurological progression of AD in a mouse model of the disease.

The experiments have been performed using primary tissue from human AD patients. For in vivo

experiments a TNFR1 mouse mutant and an established AD mouse model were used.

The majority of experiments are technically sound and the conclusions legitimate. One experiment, however, indeed the key experiment, lacks experimental stringency. Specifically, for the proof-of-concept experiment the authors used a self-developed TNFR1 function blocking nanobody termed TROS. As a control they use PBS. This is an absolutely inadequate control. The authors are advised to use a comparable unspecific nanobody as a negative control because the application of large amounts of antibodies, of any antibody indeed, is prone to side effects. Nanobodies are less well characterized than antibodies but for exactly this reason solid controls are imperative to prevent artefacts.

Further comments:

- Fig.2: In the legend the control is stated to be "not shown". This is misleading because the experimental data have been normalized to the respective controls; therefore, the controls are implicitly included in the presented data.
- Fig.5: Images b and d seem to have been mixed up.
- Fig.8: It is not clear how exactly the microglia cell count was carried out. The authors mention Volocity in the M&M part, but they should provide more detailed information on the read-out protocol including the relevant parameter settings of this software analysis tool.
- Fig.9, mislabeling in legend: STM is shown in panel (a), LTM in panel (b)

1st Revision - authors' response

15 November 2017

Referee #1 (Comments on Novelty/Model System):

The authors have been diligent in using both the APP/PS1 model and ICV injection of A β O but there are what might be considered rather extreme responses to the acute model and the disease relevance of the doses of A β O require more contextualisation. In particular in the section of choroid plexus epithelial damage, it is really not clear whether such an injury arising from an acute ICV injection is of real relevance to AD pathology.

We understand the reviewer's concern. As described below, we took these remarks into account and contextualized the discrepancies between the two models in the revised manuscript.

Referee #1 (Remarks):

In this study the Vandenbroucke group study the role of TNF signaling in various aspects of AD-related pathology. The work is generally of high quality and the authors go to some lengths to cover different approaches to studying events likely to occur in the disease: human material, APP/PS1 mice and Amyloid-beta oligomers. This is more than most groups do and the authors should be encouraged for this approach. Nonetheless, some of the key questions arise precisely because of the difficulty in comparing what happens in the 'chronic' versus the 'acute' model.

Methods

Human choroid plexus were said to be snap frozen...were they obtained directly at post-mortem. What was the delay from death to collection?

We added all the details of the sample processing in the revised manuscript as follows:

“Human choroid plexus samples of AD patients and healthy controls were obtained post-mortem at autopsy from patients with advanced late-stage Alzheimer's disease (Braak & Braak stage V-VI) and from healthy controls (mean age 58 years). Samples were immediately snap-frozen upon removal in liquid nitrogen. All samples were obtained from 3.5 to 30 h post mortem. All tissues were stored at -80°C at the Brown University Brain Tissue for Neurodegenerative Disorders Resource Center until processing.”

Figure 1

To clarify, in Figure 1a: TNF appearing as the top hit does not imply that TNF was the most highly expressed, but that a large panel of genes that share TNF-signaling as a common inducer make TNF-signaling the top hit? Likewise for IL-1 as the second hit? Figure 1b is presented at a size too small for genes to be legible.

Indeed, using Ingenuity Pathway analysis (IPA), we determined the z-score of inflammatory upstream regulators, based on in the mRNA expression in choroid plexus of Alzheimer's disease patients compared to age-matched controls. A positive z-score means that the genes downstream of

that molecule are mainly activated. In our case, *TNF* showed the highest z-score, so most of the genes downstream of *TNF* are upregulated/activated in the choroid plexus of Alzheimer's disease patients, while *IL-1 β* is the second highest upstream inflammatory regulator. As asked by the referee, we increased the size of Fig. 1B.

Figure 2.

It is appropriate that the authors note that the addition of A β O is an acute neurotoxicity model. However, the acute nature of this stimulus is very apparent in the very large responses in the choroid plexus and hippocampus compared to the APP/PS1 mice. That choroid plexus responses are 10-100 fold higher than in the APP/PS1 mice should be made explicit in the text. Moreover, the concentration of A β O injected ICV should be explicit in the results text or the legend and some attempt to compare these doses to levels of Ab oligomers in the APP/PS1 brain should be made. The magnitude of the inflammatory transcript increases seem excessively high and it highlights the need for genuine discussion of the appropriateness of acute A β O injection as a model of anything that is happening in Alzheimer's disease. Presumably these oligomers are being made all the time in the brain tissue of the AD patient and the APP/PS1 mice? Why do the hippocampus and the choroid plexus produce 2 fold and no increase respectively if these oligomers robustly induce TNF-a?

In all our experiments, oligomers were allowed to aggregate for 1.5 h before injection and mice were injected intracerebroventricularly (icv) with 5 μ l of 220 nM A β ₁₋₄₂, resulting in ~125 ng/ml assuming that there is 40 μ l total CSF. It has been previously reported that the concentration of A β in CSF of transgenic APP/PS1 mice could be as high as 30 ng/ml (A β ₄₀ and A β ₄₂ together) [1]. We have used ~4 fold higher concentration of A β ₁₋₄₂ to study the acute effects on the brain and we highlighted this more in the discussion section of the revised manuscript.

Additionally, in literature, various forms and concentrations of amyloid peptide were used *in vivo*. In these experiments, several readouts were investigated to assess the effects of amyloid peptides such as tau hyperphosphorylation, neuronal cell loss, deficits in hippocampus-dependent memory, lipid peroxidation, cholinergic neuronal degeneration, ... [2-5]. Although the concentration of soluble A β ₁₋₄₂ is not uniform throughout these studies, in most of them A β ₁₋₄₂ was able to initiate a cascade of the events that recapitulate the key pathological hallmarks of Alzheimer's disease. We included some of these studies in the introduction and discussion of the revised manuscript to stress the relevance of this model to study the amyloid-driven component of the Alzheimer's disease pathogenesis as follows (discussion):

“For the second model, we injected oligomerized A β ₁₋₄₂ (A β O) into the brain ventricles to study the direct toxic effects of A β O species. Also this model mimics key pathological characteristics of AD, such as tau hyperphosphorylation, neuronal cell loss, and impairment of hippocampus-dependent memory, making this a relevant model to study the amyloid-driven component of AD pathogenesis [2, 3, 6, 7].”

Additionally, we explicitly mentioned in the results section that the expression levels in the A β O model were much higher than these in APP/PS1 mice as follows:

“Strikingly, the fold induction of the inflammatory genes in choroid plexus and hippocampus was hundred to thousand times higher in this model of A β O-induced toxicity than in the respective brain structures of APP/PS1 mice.”

In the icv injection model, the injected A β O is the first inflammatory trigger that is encountered by the choroid plexus and the hippocampus. In contrast, the brains of APP/PS1^{tg/wt} mice are continuously triggered by the increased amyloid beta levels, typically resulting in desensitization and/or exhaustion of the activated immune cells. Consequently, the transcriptional activation of inflammatory mediators will be less pronounced. Also, *TNF* is a highly potent cytokine and even low *TNF* levels (in the picomolar range) are able to activate the *TNF* receptor and to mediate downstream effects [8]. We believe that icv injection of A β O mimics better the early phase of AD development and allows the investigation of the direct effect of A β O on the brain or on specific brain structures such as the choroid plexus.

We added this to the discussion part of the paper as follows:

“In both models, we confirmed NF- κ B-induced inflammation in the choroid plexus and hippocampus. Similar to the choroid plexus expression in late-stage AD patients, the expression of *Tnf* was not elevated in the choroid plexus of late-stage APP/PS1^{tg/wt} mice. Conversely, *Tnf* is highly expressed in both choroid plexus and hippocampus of wild type (WT) mice icv injected with A β O. Also the expression of other inflammatory genes was 100-1000 times higher after A β O injection in WT mice compared to the respective expression in APP/PS1 mice. This might be partially explained by the fact that the injected A β O is fourfold higher than the levels measured in APP/PS1^{tg/wt} mice [1]. Moreover, chronic exposure to an inflammatory trigger such as A β might exhaust and/or

desensitize inflammation-responsive cells. As APP/PS1^{tg/wt} mice are continuously exposed to toxic A β species throughout their life, this might explain the lower induction of inflammatory genes in these mice.”

Figure 3

Much of the labeling of axes is infuriatingly small even when replicated at full printed page size. The animals/treatments on the X axis are illegible. Some of these comments may suffer from misconceptions based on my inability to read the axes, but figure 3f (IL-1 in HPC) does not seem consistent with the decreased IL-1 β expression presented in Figure 2b. For the A β O, blocking TNF signaling abrogates most or all inflammatory responses.

As ask by the referee, we increased the size of the axis of all images to increase the readability.

In **Fig. 3F** of the revised manuscript, comparing hippocampus expression levels in transgenic APP/PS1 mice and non-transgenic controls, we show *Nos2* expression and this revealed no difference between APP/PS1^{tg/wt} and APP/PS1^{wt/wt} mice. This is in agreement with the graph displayed in **Fig. 2B** of the revised manuscript that also shows expression levels in APP/PS1 mice. In **Fig. 3E** of the revised manuscript, we show that *Il1 β* levels in the hippocampus of APP/PS1^{tg/wt}TNFR1^{+/+} mice are slightly although not significantly decreased compared to non-transgenic controls and this is also in agreement with the *Il1 β* levels shown in **Fig. 2B**). So we did not find any inconsistencies between both figures.

Indeed, abrogation of the TNFR1 signaling leads to a significant reduction in inflammatory expression in the choroid plexus and hippocampus for almost all genes and in the two different models.

Figure 4, 5

The effects of A β O on the choroidal epithelium are very impressive, and are prevented in TNFR1^{-/-} mice. However, the A β O dose once again becomes important to contextualize with respect to AD-associated levels of similar oligomers. The authors report morphological change of the CP in AD patients and also report changes to epithelial cell morphology in their prior A β O experiments. However, they make no attempt to compare/contrast these alterations to the CP in their acute model to any changes in AD patients and do not mention what happens to the CP in APP/PS1 mice. It would be extremely useful to show to what extent there are similar changes in the CP of APP/PS1 mice. Of course they are likely to be much less obvious and this is part of the problem: it once again raises the hard-to-ignore feeling that A β O provides a highly exaggerated version of amyloid-induced pathological changes

We agree with this concern. However, as published before and reviewed by Balusu *et al.* [9], several studies reported severe (morphological) changes at the interface of the choroid plexus. Additionally, also we are currently studying this in more detail.

To address the concern of the referee, we analyzed the choroid plexus of ~20 week old APP/PS1^{wt/wt} and APP/PS1^{tg/wt} mice in TNFR1^{+/+} and ^{-/-} background using both TEM and volume scanning electron microscopy (SEM). As shown in **Rebuttal Fig. 1 (Fig. EV1** of the revised manuscript), in agreement with the A β O effects on the choroid plexus of TNFR1^{+/+} and ^{-/-} mice, 20 week old APP/PS1^{tg/wt} mice show less damage in the choroid plexus of TNFR1^{-/-} compared to TNFR1^{-/-} background. These TEM results were also confirmed on the 3D movies obtained by SEM (**Rebuttal Fig. 2** and **Fig. EV2** of the revised manuscript). We also extensively discussed these observations in the result section of the revised manuscript as follows:

“To confirm these results in a transgenic model of AD, we dissected the choroid plexus from 18 week old APP/PS1^{tg/wt} mice in a WT and TNFR1^{-/-} background and from age-matched non-transgenic littermates. We investigated the structural alterations of the CPE cells with TEM (**Appendix Fig. EV1**) and SEM (**Appendix Fig. EV2 and Movie EV5-8**) and found that 18 week old non-transgenic littermates, *e.g.* APP/PS1^{wt/wt} mice both in WT and TNFR1^{-/-} background, already exhibited a limited degree of morphological changes (**Appendix Fig. EV1A-B**). Some of the CPE cells lost their cuboidal structure and became more point-shaped. These changes are due to ‘normal healthy’ ageing of mice, which is known to be associated with a number of ultrastructural alterations in the choroid plexus [10]. The cytoplasm and mitochondria were still preserved in mice of that age. Contrastingly, TEM images of the CPE cells of age-matched transgenic APP/PS1^{tg/wt} mice in a WT background displayed a profound transformation into cells with a more degenerative state (**Appendix Fig. EV1C**). Indeed, the cytoplasm of these cells is translucent indicative for cellular degradation and the nuclei have irregular shapes (**Appendix Fig. EV1C, zoom**). It is also clear that the capillaries of the choroid plexus are swollen and dilated, and filled with a lot of red blood cells. TNFR1 deficiency in APP/PS1^{tg/wt} mice clearly protects the morphology of the CPE cells at several

levels. There are less signs of cellular degradation, the cuboidal cell shape is as maintained as in non-transgenic littermates and the mitochondria and nuclear shape are preserved (**Appendix Fig. EV1D and zoom**). Also the capillaries are less swollen and less filled with RBCs of APP/PS1^{tg/wt} mice in a TNFR1^{-/-} background. Importantly, the microvilli of APP/PS1^{tg/wt} mice in the WT as well as the TNFR1^{-/-} background were not affected as is the case in A β O-injected WT mice, indicating that A β O species are acutely toxic for these CPE structures (**Fig. 4C-D; Appendix Fig. EV1C and D**). SEM analysis of the CPE cells of APP/PS1 mice confirmed our observations made by the TEM: CPE cells of non-transgenic APP/PS1 mice show some signs of ageing-related damage which is much more pronounced in a transgenic background, whereas CPE cells of APP/PS1^{tg/wt}TNFR1^{-/-} mice are kept intact (**Fig. EV2A-D, Appendix Movie EV5-8**).”

Rebuttal Figure 1

TNFR1 deficiency protects against morphological alterations in the choroid plexus of APP/PS1^{tg/wt} mice determined by TEM

A-D Representative conventional transmission electron microscopy (TEM) images of the choroid plexus of 18 week old C57BL/6J APP/PS1^{tg/wt} mice in a TNFR1^{+/+} and TNFR1^{-/-} background compared to age-matched non-transgenic controls. In non-transgenic controls (**A,B**), the cuboidal structure of the choroid plexus epithelial (CPE) cells is already a bit affected because of normal healthy ageing. The nuclei have regular shapes and mitochondria look normal (*zoom*). The loss of cuboidal shape is more enhanced in CPE cells of APP/PS1^{tg/wt} TNFR1^{+/+} mice (**C**). The capillaries are swollen and filled with plenty of red blood cells, the nuclei have irregular shapes and some CPE cells are at a degenerative state (*zoom*). In contrast (**D**), CPE cells of APP/PS1^{tg/wt}TNFR1^{-/-} mice have the some cellular shape as non-transgenic littermates. The capillaries are less swollen, and the mitochondria and nuclear shape are normal (*zoom*).

Data information: The TEM images were taken at a magnification of 1000x, scale bar represents 10 μ m; zooms were taken at a magnification of 3000x, scale bar represents 2 μ m.

Rebuttal Figure 2

TNFR1 deficiency protects morphological alterations in the choroid plexus of APP/PS1^{tg/wt} mice determined by SBF-SEM

A-D Representative serial block-face scanning electron microscopy (SBF-SEM) images of the choroid plexus of 18 week old C57BL/6J APP/PS1^{tg/wt} mice in a TNFR1^{+/+} (A,C) and TNFR1^{-/-} (B,D) background compared to age-matched non-transgenic controls (n=1/group), derived from *Movie EV1-4*.

Data information: CSF: Cerebrospinal fluid; Mv, Microvilli; Nu, Nucleus; Scale bar, 5 μm.

Figure 6

Increased dextran leakage into the CSF occurs after AβO and this should be more explicit in the way that Fig 6a is labeled (on the axes) or explained (in the legend). The reader should not have to return to the methods section to interpret what this graph means.

To address this, we clearly stated how blood-CSF barrier permeability was determined in the legend of the figures in the revised manuscript.

In (f) the labels are, once again, hard to read. It looks, to this reader as though the CLDN1 tight junctions are more intact in the AβO TNFR1^{-/-} mice than in the scrambled group. I do see that the pictures are annotated with arrows and arrowheads but the patterns look much more similar between the left and centre than between the left and right. How robust is this measure? Even the areas chosen to compare are quite different (ie far less CP in the left image).

As mentioned earlier, we increased the label size in different images throughout the manuscript to increase the readability.

We agree with the comment related to the CLDN1 stainings. We revised our figure and agree that the selected images of the scrambled WT mice might not be ideal. However, we can assure that the

effect on CLDN1 localization is a very robust effect. By selecting a new image, we now show a similar region in all conditions and this clearly shows that upon A β O injection in WT mice, CLDN1 is located much more in the cytoplasm compared to the scrambled injected condition and in absence of TNFR1 (Rebuttal Fig. 3F).

Rebuttal Figure 3

TNFR1 deficiency protects the blood–cerebrospinal fluid (CSF) barrier by preventing MMP increase and preserving tight junctions

A Blood–CSF barrier permeability was determined by measuring leakage of FITC-labeled dextran in the CSF of C57BL/6J TNFR1^{+/+} and TNFR1^{-/-} mice 6 h after intracerebroventricular (icv) injection of either scrambled peptide or A β ₁₋₄₂ oligomer (A β O) (n = 11-14/group).

B-E Relative mRNA gene expression of *Mmp8*, *Mmp3*, *Cldn5* and *Ocln* in choroid plexus of TNFR1^{+/+} and TNFR1^{-/-} mice 6 h after icv injection of A β O (n = 5-6/group).

F Representative images of CLDN1 staining in choroid plexus of the fourth ventricle of TNFR1^{+/+} (left and middle image) and TNFR1^{-/-} mice (right image) 6 h after icv injection with either 1 μ g/ml A β O (middle and right image) or scrambled peptide (left image). Arrows indicate preserved CLDN1 tight junctions and arrowheads indicate affected CLDN1 tight junctions (n = 3/group).

Data information: Bars represent mean \pm SEM. qPCR was normalized to stable housekeeping genes determined by GeNorm. Scale bar represents 15 μ m. The experiments are done in duplicates and pooled results are shown in (A) and representative results in (B-E). Statistics were performed with a one-way ANOVA for the permeability data (A) and an unpaired t test for the qPCRs (B-E), * 0.01 \leq p < 0.05; ** 0.001 \leq p < 0.01; *** 0.001 \leq p < 0.0001, **** p < 0.0001

Figure 7

Several graphs of amyloid quantification are presented but none of the raw data are visible. For example pictures of thioflavin S (ThioS) labeling should be shown in relevant experimental groups. The text reflects 'significantly fewer' plaques of various sizes but this refers to statistical significance: they should also reflect the fairly modest decreases (i.e. 15-20% reduction). The statistics that have been carried out are only partially covered in the figure legend, with no information for c and d and a descriptor suggesting that there are panels e-j (obviously a relic of a prior version).

We agree that the decrease is fairly modest in the $TNFR1^{-/-}$ background, and therefore, the difference between the two groups is difficult to appreciate visually. Nevertheless, to meet the reviewer's demand, we added a representative image of the brain of each group, as shown in **Rebuttal Fig. 4C-D** (**Fig. 7C-D** in the revised manuscript). In the result section, we clearly stated that this difference is significant but limited.

We also critically revised the figure legend of **Fig. 7** of the original manuscript and we corrected the mistakes in the numbering and in annotation of the statistics.

Rebuttal Figure 4

TNFR1 deficiency reduces Aβ pathology and prevents cognitive decline in APP/PS1 mice

A-D Brain sections of late-stage C57BL/6J APP/PS1^{tg/wt} and APP/PS1^{tg/wt}TNFR1^{-/-} mice were stained with Thioflavin-S to detect Aβ disposition in the whole brain. The amount of plaques was quantified (**A**) and a morphometric analysis was performed (**B**). (n = 5 and 6 mice/group, respectively). Representative images (**C-D**) of Thioflavin-S (ThioS) staining of the brain containing the hippocampus of late-stage APP/PS1^{tg/wt} and age-matched APP/PS1^{tg/wt}TNFR1^{-/-} mice (scale bar represents 200 μm).

E-H ELISA analysis of soluble and insoluble Aβ₁₋₄₀ and Aβ₁₋₄₂ in the cortex of late-stage APP/PS1^{tg/wt} in a TNFR1^{+/+} and TNFR1^{-/-} background compared to age-matched controls (n = 8, 9 and 3 mice, respectively).

I Relative mRNA expression of *Bace1* in the hippocampus of late-stage APP/PS1^{tg/wt} mice in a TNFR1^{+/+} and TNFR1^{-/-} background compared to age-matched APP/PS1^{wt/wt} mice (n = 5/group).

Data information: Bars represent mean ± SEM. qPCR was normalized to stable housekeeping genes determined by GeNorm. Statistics were performed with a one-way ANOVA (**A,E-H**), a two-way ANOVA assay (**B**) or an unpaired t test (**I**), * 0.01 ≤ p < 0.05; ** 0.001 ≤ p < 0.01; *** 0.001 ≤ p < 0.0001, **** p < 0.0001

It is not clear why cognitive impairments in the APP/PS1 model are tagged on here but cognitive deficits in the AβO model are tagged onto IBA-1 labeling that is shown for both models.

We critically revised the manuscript and changed the order of the figures to make the build-up of the story more logical. We merged all figures about the cognitive deficits and generated one new figure (**Fig. 9A-F** of the revised manuscript). Now, **Fig. 7** of the revised manuscript only focuses on the Aβ load and **Fig. 8** of the revised manuscript focuses on the involvement of microglia in the two models. We also rearranged the text of the manuscript to guide the reader through the manuscript in an obvious way.

If I understand the bar and min-max box plots, the values annotated and included in the text do not look right: In fig 7i the left bar is said to be 64.18 but the bar looks to be below 60, while in the middle bar, it is said to be 50.8 but the bar looks more like 55. This requires some explanation (and is of pressing relevance to key result).

We critically revised all box plots shown in the manuscript (also those displayed in **Fig. 8 and 9**) and all values mentioned both in the text and in the figures are correct in the revised manuscript. It might have been confusing in the original manuscript, since the mean value is displayed above the box plots, while the horizontal line in the box plot itself indicates the median value. We clarified this in all figure legends to avoid confusion.

Figure 8

The IBA-1 pictures are inadequate to show what the graphs refer to. Is brown staining really quantified in the 'whole brain section'? Why would the authors choose to image 'whole brain section': why not representative areas in the hippocampus and cortex so that cells can be examined to analyze the pattern of microgliosis within these areas? If the images in c represent an 'area close to the choroid plexus', why is the choroid plexus not shown and why does the middle image not contain the (presumed) ventricle that is visible in the other 2 pictures? The data presented in the graphs look very robust and the images to support this really should be shown. For quantification of images to be credible the reader should be able to examine sample images to increase confidence in the numbers generated from these images.

We quantified the IBA-1 staining on whole brain sections of A β O-injected mice, while the image indeed shows the region close to the ventricles. In the revised manuscript, we made sure that the ventricular space (containing the choroid plexus) is always visible. As correctly stated by the referee, we don't show the choroid plexus, so we changed the statement in the figure legend from 'an area close to the choroid plexus' to 'an area close to the ventricles'.

We selected this region, because the A β O is injected into the brain ventricles and therefore, we expect a high inflammatory response in the brain parenchyma close to the ventricles. However, the difference in microgliosis is observed equally considering all regions of the brain. Indeed, quantification of the % brownstaining (determined by both amount and size of the IBA-1 positive cells) in hippocampus or in the regions surrounding the ventricles was only slightly (non-significantly) higher in WT mice compared to TNFR1^{-/-} mice. In contrast, analysis of whole brain sections shows a significantly lower IBA-1 positive signal in TNFR1^{-/-} mice.

Why are pictures shown for the A β O-injected but not the APP/PS1? What is the quality of the IBA-1 labelling? With such a clear effect of TNFR1 deficiency on microglial-positive area one would really like to see this in some pictures.

We agree with the remark of the reviewer about the missing IBA-1 staining in APP/PS1 mice and therefore we also added representative brain images of late-stage APP/PS1^{tg/wt}TNFR1^{+/+} mice compared with non-transgenic littermates, and we compared this with brain images of APP/PS1^{tg/wt} and APP/PS1^{tg/wt} mice in TNFR1^{-/-} background (**Rebuttal Fig. 5A-B** and **Fig. 8A-B** in the revised manuscript).

Rebuttal Figure 5

TNFR1 deficiency prevents microglia activation in APP/PS1 mice and upon icv AβO injection

A-B IBA1 staining for microglia on whole-brain sections of late-stage C57BL/6J APP/PS1^{tg/wt} in a TNFR1^{+/+} and TNFR1^{-/-} background compared to age-matched controls. **(A)** Representative images of a region around the fourth ventricle (microglia indicated with *arrowheads*) in age-matched controls (*upper panel*, n = 3), APP/PS1^{tg/wt}TNFR1^{+/+} (*middle panel*, n = 4) and APP/PS1^{tg/wt}TNFR1^{-/-} (*lower panel*, n = 6) mice. **(B)** Quantification of IBA1⁺ cell count (determined by brown staining).

C-D IBA1 staining for microglia on whole-brain sections of C57BL/6J TNFR1^{+/+} mice 6 h after intracerebroventricular (icv) injection with either scrambled peptide or Aβ₁₋₄₂ oligomer (AβO, 1 μg/ml) or TNFR1^{-/-} mice icv injected with AβO. **(C)** Representative images of a region around the fourth ventricle (microglia indicated with *arrowheads*) in scrambled-injected mice (*upper panel*, n = 2) and in AβO-injected TNFR1^{+/+} (*middle panel*, n = 3) and TNFR1^{-/-} mice (*lower panel*, n = 3) **(D)** Quantification of IBA1⁺ cell count (determined by brown staining).

Data information: Scale bars represent 100 μm and insert 20 μm Bars represent mean ± SEM. Statistics were performed with a one-way ANOVA assay, * 0.01 ≤ p < 0.05; ** 0.001 ≤ p < 0.01; *** 0.001 ≤ p < 0.0001, **** p < 0.0001

As highlighted for Fig. 7I above, in Fig. 8E the middle bar is said to be 49.99 but the bar looks to be more like 53 to 54. Again, this requires some explanation and is relevant for the key result. There are some similar issues to address in Fig. 9.

As explained above, we critically revised all the box plots shown in the manuscript (Figure 7, 8 and 9) and clarified the meaning of the values and box plots.

The latter part of the paper feels a little disorganized, with the cognitive data distributed in a way that does not make obvious sense to this reader. I think it could be significantly improved by taking time to present the amyloid more clearly and with pictures, in a figure on its own. Then doing likewise for IBA-1, again showing pictures in both models. Finally the NOR task could be presented for all three models in a third figure - but with more attention to the detail of the box plots.

We took the suggestion of the reviewer into account and we significantly changed the order of the revised manuscript to improve the structure and comprehensibility. We want to thank the reviewer for the suggestion.

I would also wonder why, after so much focus on the choroid plexus early in the paper, there is not more attention on implicating the choroid plexus changes in the cognitive impairments ultimately observed. It gives a sense of discontinuity despite the impressive effects of TNF disruption on almost every measure examined. Which ones are most important for cognitive dysfunction is a key question and whether that parameter would stand up across the multiple models would obviously be important to establish.

This is a very interesting question and this is for sure something we want to address in the future. We are currently generating inducible, choroid plexus epithelial specific Cre mice. We will, if successful, cross the mice with the available TNFR1^{fl/fl} mice. Next to this strategy, we want to stress that, despite available data in literature we were not successful in e.g. delivering high levels of recombinant Tat-Cre in the choroid plexus epithelial cells upon injection in the ventricles of TNFR1^{fl/fl} mice to obtain choroid plexus specific TNFR1 deficient mice. Additionally, the two choroid plexus specific Cre mice described in literature were tested, but were either not specific or not breeding properly.

Referee #2 (Comments on Novelty/Model System):

The key experiment lacks an adequate control, see statement below. Other than that, the technical quality of the work is good.

Referee #2 (Remarks):

In the present manuscript Steeland et al. report on signaling activity of TNFR1 that might contribute to manifestation of Alzheimer's disease (AD). Interfering with this signaling activity appears to attenuate the pathophysiological and neurological progression of AD in a mouse model of the disease.

The experiments have been performed using primary tissue from human AD patients. For in vivo experiments a TNFR1 mouse mutant and an established AD mouse model were used.

The majority of experiments are technically sound and the conclusions legitimate. One experiment, however, indeed the key experiment, lacks experimental stringency. Specifically, for the proof-of-concept experiment the authors used a self-developed TNFR1 function blocking nanobody termed TROS. As a control they use PBS. This is an absolutely inadequate control. The authors are advised to use a comparable unspecific nanobody as a negative control because the application of large amounts of antibodies, of any antibody indeed, is prone to side effects. Nanobodies are less well characterized than antibodies but for exactly this reason solid controls are imperative to prevent artefacts.

We understand the concern of the reviewer that we didn't take a control Nanobody along in our experiments. In studies using therapeutic antibodies, isotype control antibodies are used to correct for effects of the non-variable domain of the antibody. However, Nanobodies don't contain a variable domain, making it very difficult to have a proper control. Hence, we took advantage of the fact that our TROS Nb only recognizes human TNFR1; the reason why we used our humanized transgenic mice (hTNFR1 Tg) in our proof-of-concept study [11, 12]. Consequently, TROS treatment in wild type mice will not be able to block TNFR1 signaling and if effects are observed, this is due to the anti-albumin blockage or other non-specific effects of TROS. To address possible side effects of TROS on the A β O-induced inflammation, we injected WT mice with A β O in the presence or absence of TROS and compared this with scrambled injected mice. Six hours later, choroid plexus inflammation was analyzed by qPCR. As shown in **Rebuttal Fig. 6 (Fig. EV3)** of the

revised manuscript), TROS has no effect on inflammatory genes in the absence of its target hTNFR1, so we can assure that TROS blocks the effects of A β O by blocking TNF/TNFR1 signaling specifically. We added these data to the result section of the manuscript as follows:

“To exclude any TROS or Nanobody-related effect on inflammation upon icv injection, we administered TROS together with A β O via icv injection in WT mice, which don't express the therapeutic target of TROS, namely *hTNFR1*. We determined the expression of several inflammatory genes, e.g. *Il6*, *Il1 β* , *Tnf*, *Mmp3* and *Tnfrsf1a* in the choroid plexus upon A β O injection together with PBS and compared it with A β O/TROS co-injection. These results confirm that TROS nor increases nor reduces A β O-induced inflammation in mice that are irresponsive to the effects of TROS (Fig. EV3). Next, we assessed the memory performance after TROS treatment in hTNFR1 Tg mice, having the target of TROS.”

Rebuttal Figure 6

Icv injection of TROS in WT mice does not affect inflammation induced by A β ₁₋₄₂ oligomers (A β O)

A-E Relative mRNA gene expression of *Il1 β* , *Tnf*, *Il6*, *Mmp3* and *Tnfrsf1a* of C57BL/6J wild type (WT) mice 6 h after intracerebroventricular (icv) injection with scrambled peptide or with A β O (1 μ g/ml) together with PBS or with TROS (n = 6/group).

Data information: Bars represent mean \pm SEM. qPCR was normalized to stable housekeeping genes determined by GeNorm. Statistics were performed with an unpaired t test, * 0.01 \leq p < 0.05; ** 0.001 \leq p < 0.01; *** 0.001 \leq p < 0.0001, **** p < 0.0001

Further comments:

Fig.2: In the legend the control is stated to be "not shown". This is misleading because the experimental data have been normalized to the respective controls; therefore, the controls are implicitly included in the presented data.

We agree with the reviewer and we removed this in the legend.

Fig.5: Images b and d seem to have been mixed up.

We checked this mistake but we do not think that these were mixed-up.

Fig. 8: It is not clear how exactly the microglia cell count was carried out. The authors mention Volocity in the M&M part, but they should provide more detailed information on the read-out protocol including the relevant parameter settings of this software analysis tool.

In the original manuscript, the stainings were visualized using Olympus BX51 microscope and we analyzed and quantified the amount of IBA-1 positive cells with Volocity. In the revised manuscript, we added new images of the IBA-1 staining of TNFR1^{-/-} and TNFR1^{+/+} mice icv injected with A β O, as well as IBA-1 staining on brains of APP/PS1^{wt/tg} mice in a TNFR1^{+/+} and TNFR1^{-/-} background. The imaging of whole brain sections of these new images was performed using Zeiss Axio Scan.Z1 followed by quantification by Fiji (<http://fiji.sc/Fiji>). Quantification of the amount of brown colour

was done with colour thresholding, with correction for the total amount of brain tissue. We also added this to the revised manuscript.

Fig. 9, mislabeling in legend: STM is shown in panel (a), LTM in panel (b).

We agree with the reviewer and we corrected the labeling in the legend of this figure.

References

1. Maia, L.F.*et al.*, (2013) Changes in amyloid-beta and Tau in the cerebrospinal fluid of transgenic mice overexpressing amyloid precursor protein. *Sci Transl Med* 5, 194re2.
2. Brouillette, J.*et al.*, (2012) Neurotoxicity and memory deficits induced by soluble low-molecular-weight amyloid-beta1-42 oligomers are revealed in vivo by using a novel animal model. *J Neurosci* 32, 7852-61.
3. Cetin, F.*et al.*, (2013) The effect of intracerebroventricular injection of beta amyloid peptide (1-42) on caspase-3 activity, lipid peroxidation, nitric oxide and NOS expression in young adult and aged rat brain. *Turk Neurosurg* 23, 144-50.
4. Nitta, A.*et al.*, (1997) Continuous infusion of beta-amyloid protein into the rat cerebral ventricle induces learning impairment and neuronal and morphological degeneration. *Japanese journal of pharmacology* 73, 51-57.
5. Kong, L.-n.N.*et al.*, (2005) Gene expression profile of amyloid beta protein-injected mouse model for Alzheimer disease. *Acta pharmacologica Sinica* 26, 666-672.
6. Balducci, C. and Forloni, G., (2014) In vivo application of beta amyloid oligomers: a simple tool to evaluate mechanisms of action and new therapeutic approaches. *Curr Pharm Des* 20, 2491-505.
7. Nitta, A.*et al.*, (1997) Continuous infusion of beta-amyloid protein into the rat cerebral ventricle induces learning impairment and neuronal and morphological degeneration. *Jpn J Pharmacol* 73, 51-7.
8. Grell, M.*et al.*, (1995) The transmembrane form of tumor necrosis factor is the prime activating ligand of the 80 kDa tumor necrosis factor receptor. *Cell* 83, 793-802.
9. Balusu, S.*et al.*, (2016) The choroid plexus-cerebrospinal fluid interface in Alzheimer's disease: more than just a barrier. *Neural Regeneration Research* 11, 534-7.
10. Gorle, N.*et al.*, (2016) The effect of aging on brain barriers and the consequences for Alzheimer's disease development. *Mamm Genome* 27, 407-20.
11. Steeland, S.*et al.*, (2015) Generation and characterization of small single domain antibodies inhibiting human tumor necrosis factor receptor 1. *J Biol Chem* 290, 4022-37.
12. Steeland, S.*et al.*, (2017) TNFR1 inhibition with a Nanobody protects against EAE development in mice. *Sci Rep* 7, 13646.

2nd Editorial Decision

12 December 2017

Thank you for the submission of your manuscript to EMBO Molecular Medicine. You will see that while referee #2 is now satisfied, referee #1 still finds some aspects of the paper that were not adequately addressed and we would like to encourage you to address these as soon as possible, in light of this referee's recommendations.

We would welcome the submission of a revised version for further consideration and depending on the nature of the revisions, this may be sent back to referee #1 for another round of review.

I look forward to seeing a revised form of your manuscript as soon as possible.

***** Reviewer's comments *****

Referee #1 (Remarks for Author):

The authors have made genuine attempts to address all of my concerns. However I there are some issues that remain.

1) The authors have made several references to the differences between the 2 model systems. Broadly speaking these are reasonable.

2) Methods:

Human choroid plexus: Choroid plexus was sourced from AD patients and healthy aged individuals (mean age 58). One assumes that the average age of the AD choroid plexus samples was older than 58 and that the healthy controls, therefore are not age-matched. This should be specified and appropriate caveats added if necessary.

3) z-scores in ingenuity analysis and size of figure 1: Fine.

4) Figure 2: the discussion of the nuances of acute A β O versus slow build up of amyloid is helpful.

5) Figure 3: labelling is clearer. I have no problems with the IL-1 and iNOS data shown in figs 2 and 3. However I am only now noticing that several graphs appear to be reliant on n=2 samples which limits the credibility of any statistical analyses of those data.

6) Figure 4,5

The authors state (and show) that there are indeed changes in the choroid plexus of APP/PS1 mice using TEM analysis. They provide many images that will not be available to the reader of any eventual manuscript (since they are being prepared for another manuscript), which increases the pressure to present some images that will be available to all readers. Navigating this part of the rebuttal is not helped by the fact that the actual figures in the PDF manuscript are not labelled with figure 1, 2, 3 etc. I am assuming that the last 3 figures of the PDF are EV1, EV2, EV3? If this is the case and all three figures will be visible to the readers then this is probably satisfactory.

However, I don't imagine it would significantly compromise a future detailed investigation of the CP in AD and APP/PS1 mice to show here, in the main manuscript, that significantly altered morphology of the choroid plexus is not particular to treatment with A β O but rather is also observed in APP/PS1 mice and influenced by TNFR1 expression. A single time point of APP/PS1 versus age matched control should be sufficient to convince the readers that morphological changes in the choroid plexus occur whether during acute or chronic exposure to A β , but these changes should be clearly signalled to the reader and ideally should be placed beside and compared directly with those changes observed after treatment with A β O.

With respect to what IS shown, it is surprising to read of changes associated with "normal healthy aging" already becoming apparent at 18 weeks of age. Can this be correct? These animals are not 'aged' by any conventional description of the term aging.

Also, in EV2D, the microvilli do not look intact in the way they appear in A,B and C of that figure and in the way they are described in the new text. What is the intense black labelling at the extremities of the microvilli that is not apparent in other images? Given that this is Tg crossed with the TNFR1-/- this is a confusing image to accommodate into the narrative and needs some explanation.

7) Figure 6

It is unhelpful to have a "rebuttal figure 4" and a manuscript "figure 6" that are identical. This tendency to have the same figure appear in 2 places (ie rebuttal and manuscript) with 2 different numbers has been used throughout the rebuttal (rebuttal figures 2 and 3 appear are identical to EV1 and EV2, rebuttal figure 5 = manuscript figure 7 etc).

The altered distribution of claudin 1 labelling that I referred to is more convincing and better labelled in the revised image (in rebuttal figure 4 and in manuscript figure 6!) but I still think that the phrasing 'located much more in the cytoplasm' is hard to support according to the images i.e. there is strong cytoplasmic labelling with claudin 1 in all images. Perhaps the authors might say less obviously apical or something else that conveys the discontinuity of cldn1 at the epithelial surface.

8) Figure 7. Images of Thioflavin S - fine. I do think that the reorganization of this section of the paper into separate A β , microglial and cognitive figures improves the flow of the paper.

Statistics in figure legend still don't look quite right: "Statistics were performed with a one-way ANOVA (A,E-H), a two-way ANOVA assay (B) or an unpaired t test (I), * 0.01 {less than or equal to} p <0.05; ** 0.001 {less than or equal to} p < 0.01; *** 0.001 {less than or equal to} p 0.0001, **** p < 0.0001". A should likely be a t-test, and I cannot be a t-test, containing, as it does, three groups. This figure legend title also says that cognitive decline is a part of this figure which it no longer is.

9) Figure 8

Quantification of microglia. I did request that microglial images should be provided and the authors have obliged. Unfortunately, the microglial labeling of some slides (including A β O, TNFR1 $^{+/+}$ and APP/PS1 Tg/wt TNFR1 $^{-/-}$) show an extremely brown pattern of labeling, staining much more than just microglia. If the method of quantification is % brown staining then one simply cannot reliably quantify the microglial response from sections like these since, if these images are representative, there is a high likelihood of overestimation of the response to A β O and the role of TNFR1. Contrast those images to APP/PS1 tg/wt in TNF1 $^{+/+}$ to see high quality labeling. This requires resolution or removal.

10) Figure 9. The cognitive data seem ok to me.

Referee #2 (Remarks for Author):

The authors have adequately addressed my concerns. The manuscript has been significantly improved in technical quality and clarity and in my view is now suitable for publication.

2nd Revision - authors' response

31 December 2017

Referee #1

Methods: Human choroid plexus: Choroid plexus was sourced from AD patients and healthy aged individuals (mean age 58). One assumes that the average age of the AD choroid plexus samples was older than 58 and that the healthy controls, therefore are not age-matched. This should be specified and appropriate caveats added if necessary.

The average age of the AD patients was 79 years (95% CI [72.2 – 86.4]) and thus is indeed slightly higher compared to our healthy controls (58 years, 95% CI [41.9 – 72.9]). We agree that this is a limitation in our human study. However, we could confirm the involvement of TNF in all our mouse models in which we used age-matched mice and this thus strengthens that the observations in human subjects are due to Alzheimer's disease and not due to ageing. Additionally, the corresponding mouse ages of the AD patients and healthy controls are 2 years (104 weeks) and 1 year (52 weeks), respectively. All our Alzheimer's research has been done in mice that were not older than 40 weeks. As this is younger than the corresponding mouse age of the healthy control group, we believe that we are effectively looking to Alzheimer's effects in the human samples and not at ageing effects. We made this more clear in the revised manuscript.

Figure 3: Labelling is clearer. I have no problems with the IL-1 and iNOS data shown in figs 2 and 3. However I am only now noticing that several graphs appear to be reliant on n=2 samples which limits the credibility of any statistical analyses of those data.

We understand the concern of this reviewer, however, we believe that this does not influence the conclusions from our manuscript; as this is only the case in a limited amount of values. Moreover, we clearly indicated the n-values in all graphs and we also supplied a table with all n- and p-values of the manuscript (**Appendix Table S2**). The statistical test that was applied on these data was fitted for its purpose and allowed correct statistical comparison of the two datasets

6) Figure 4,5: The authors state (and show) that there are indeed changes in the choroid plexus of APP/PS1 mice using TEM analysis. They provide many images that will not be available to the reader of any eventual manuscript (since they are being prepared for another manuscript), which increases the pressure to present some images that will be available to all readers.

Indeed, we are preparing another manuscript that will focus on the morphological changes at the choroid plexus at different ages in APP/PS1 mice and other Alzheimer's disease mouse models. Therefore, the APP/PS1 TEM images of 10, 20, 30 and 40 weeks old mice will not be included in the current manuscript. Importantly, the images we showed in the previous rebuttal letter are not necessary to understand the current manuscript as we did include the TEM morphological images of

age-matched APP/PS1 WT mice which allows comparison with APP/PS1 mice in a TNFR1^{-/-} background (**Fig. EV1**).

Navigating this part of the rebuttal is not helped by the fact that the actual figures in the PDF manuscript are not labelled with figure 1, 2, 3 etc. I am assuming that the last 3 figures of the PDF are EV1, EV2, EV3? If this is the case and all three figures will be visible to the readers then this is probably satisfactory.

We are aware of this problem, but this is beyond our control and inherent to the on line EMBO Mol Med system. The extended view figures have to be uploaded separately and apparently, upon pdf merging, the EV figures are added after the main manuscript, after the legends of the extended view figures without label. But as you correctly mentioned, the three last figures of the PDF are EV1, EV2 and EV3 and these data will be visible for the readers as described in the rebuttal letter.

However, I don't imagine it would significantly compromise a future detailed investigation of the CP in AD and APP/PS1 mice to show here, in the main manuscript, that significantly altered morphology of the choroid plexus is not particular to treatment with A β O but rather is also observed in APP/PS1 mice and influenced by TNFR1 expression. A single time point of APP/PS1 versus age matched control should be sufficient to convince the readers that morphological changes in the choroid plexus occur whether during acute or chronic exposure to A β , but these changes should be clearly signalled to the reader and ideally should be placed beside and compared directly with those changes observed after treatment with A β O.

As indicated above, we indeed included the appropriate APP/PS1 WT and TNFR1^{-/-} TEM images, next to TEM images of the A β O-treated WT and TNFR1^{-/-} mice as extended view figures. As discussed in the manuscript, CP morphology is altered in the two models, indicating that both an acute A β O insult and chronic amyloidosis induce inflammation and leads to morphological alterations. These effects are TNFR1-dependent in the two models. We clearly stated this in the manuscript as we first show the effects of acute A β O exposure on the CP morphology and the preservation of the CP morphology in the TNFR1^{-/-} mice and then we confirm these observations in the chronic APP/PS1 model – that obviously also involves the presence of A β O but in lower concentrations – in the two genetic backgrounds.

With respect to what IS shown, it is surprising to read of changes associated with "normal healthy aging" already becoming apparent at 18 weeks of age. Can this be correct? These animals are not 'aged' by any conventional description of the term aging.

Indeed, there is barely any structural damage visible at 18 weeks of age. To avoid confusion, we adapting the sentence in the revised manuscript.

Also, in EV2D, the microvilli do not look intact in the way they appear in A,B and C of that figure and in the way they are described in the new text. What is the intense black labelling at the extremities of the microvilli that is not apparent in other images? Given that this is Tg crossed with the TNFR1^{-/-} this is a confusing image to accommodate into the narrative and needs some explanation.

We partially agree with this concern. Unfortunately, the black labelling found at the edges of the microvilli in **Fig. EV2D** are due to a technical artefact in the APP/PS1^{tg/wt} TNFR1^{-/-} SEM sample, most likely caused by osmium precipitation that occurred at one of the processing steps. However, the microvilli are not affected; the precipitation is only present at the outside of the sample. Importantly, this precipitation is not visible in the TEM images of the same mice – TEM and SEM images are made from CP taken from the same mouse – and we can assure that this has no impact on our conclusions. In the text, we only refer to the TEM images of these mice when stating that the microvilli are preserved as these images clearly show that the microvilli remained intact in the APP/PS1^{tg/wt} TNFR1^{-/-} mice, compared to their wild type counterparts.

Figure 6: *It is unhelpful to have a "rebuttal figure 4" and a manuscript "figure 6" that are identical. This tendency to have the same figure appear in 2 places (ie rebuttal and manuscript) with 2*

different numbers has been used throughout the rebuttal (rebuttal figures 2 and 3 appear are identical to EV1 and EV2, rebuttal figure 5 = manuscript figure 7 etc).

This was done on purpose to allow the reviewer to read the rebuttal independently from the manuscript. In the rebuttal letter, we numbered the figures according to their appearance in the rebuttal text. Additionally, we always referred to the corresponding figure in the manuscript as the rebuttal letter and the manuscript are two different files and this might facilitate reading the two files separately. We want to apologize if this has generated some confusion.

The altered distribution of claudin 1 labelling that I referred to is more convincing and better labelled in the revised image (in rebuttal figure 4 and in manuscript figure 6!) but I still think that the phrasing 'located much more in the cytoplasm' is hard to support according to the images i.e. there is strong cytoplasmic labelling with claudin 1 in all images. Perhaps the authors might say less obviously apical or something else that conveys the discontinuity of cldn1 at the epithelial surface.

As suggested by the reviewer, we rephrased this sentence as follows:

“Six hours after A β O injection, CLDN1 remained apical in the choroid plexus epithelial cells of injected TNFR1^{-/-} mice (**Fig. 6F, arrows**) whereas in WT mice it became more diffuse, its expression weakened, and is less obviously prominent in the apical region (**Fig. 6F, arrowheads**).“

Figure 7: *Statistics in figure legend still don't look quite right: "Statistics were performed with a one-way ANOVA (A,E-H), a two-way ANOVA assay (B) or an unpaired t test (I), * 0.01 {less than or equal to} p < 0.05; ** 0.001 {less than or equal to} p < 0.01; *** 0.001 {less than or equal to} p 0.0001, **** p < 0.0001". A should likely be a t-test, and I cannot be a t-test, containing, as it does, three groups. This figure legend title also says that cognitive decline is a part of this figure which it no longer is.*

We apologize for this mistake and corrected it in the revised manuscript.

Figure 8: *Quantification of microglia. I did request that microglial images should be provided and the authors have obliged. Unfortunately, the microglial labeling of some slides (including A β O, TNFR1^{+/+} and APP/PS1 Tg/wt TNFR1^{-/-}) show an extremely brown pattern of labeling, staining much more than just microglia. If the method of quantification is % brown staining then one simply cannot reliably quantify the microglial response from sections like these since, if these images are representative, there is a high likelihood of overestimation of the response to A β O and the role of TNFR1. Contrast those images to APP/PS1 tg/wt in TNFR1^{+/+} to see high quality labeling. This requires resolution or removal.*

We do not agree with this remark. For the analysis and quantification of the IBA1⁺ cells, we thresholded the brown staining for each sample and we strictly controlled whether only the IBA⁺ brownstaining was considered without taking the background staining into account. Therefore, we can assure that the correct measurements are made. We also stated this in the material and methods section of the manuscript.

Thank you for the submission of your revised manuscript to EMBO Molecular Medicine. We have now re-assessed it and are pleased to inform you that we will be able to accept it pending following a few final editorial amendments.

Corresponding Author Name: Roosmarijn Vandenbroucke

Manuscript Number: EMM-2017-08300